# Rheological inheritance controls the formation of segmented rifted margins in cratonic lithosphere

M. Gouiza [1][✉] & J. Naliboff[2,3][✉]

Observations from rifted margins reveal that significant structural and crustal variability develops through the process of continental extension and breakup. While a clear link exists between distinct margin structural domains and specific phases of rifting, the origin of strong segmentation along the length of margins remains relatively ambiguous and may reflect multiple competing factors. Given that rifting frequently initiates on heterogenous basements with a complex tectonic history, the role of structural inheritance and shear zone reactivation is frequently examined. However, the link between large-scale variations in lithospheric structure and rheology and 3-D rifted margin geometries remains relatively unconstrained. Here, we use 3-D thermo-mechanical simulations of continental rifting, constrained by observations from the Labrador Sea, to unravel the effects of inherited variable lithospheric properties on margin segmentation. The modelling results demonstrate that variations in the initial crustal and lithospheric thickness, composition, and rheology produce sharp gradients in rifted margin width, the timing of breakup and its magmatic budget, leading to strong margin segmentation.

---

[1] School of Earth and Environment, University of Leeds, Leeds, UK. [2] Department of Earth and Planetary Sciences, University of California, Davis, CA, USA. [3] New Mexico Institute of Mining and Technology, Socorro, NM, USA. [✉]email: m.gouiza@leeds.ac.uk; john.naliboff@nmt.edu

  1

The formation of rifted continental margins occurs through multiple phases of extension with distinct structural, sedimentary, and magmatic characteristics[1,2]. A synthesis of key features at Atlantic rifted margins[2] suggests that most rifted margins undergo a similar sequence of deformation phases, which reflect progressive thinning of the continental lithosphere that produces a transition from distributed to highly localized deformation. While this sequence produces genetically similar 'domains' from the un-rifted continent to the seafloor (i.e., proximal, necking, and distal domains), significant segmentation can occur either at plate-scale (e.g. South, Central, and North Atlantic segments[3]), between different rifted margins (e.g. between the Labrador Sea and the Baffin Bay through the Davis Strait[4]), or within a single basin (e.g. Labrador Sea[5]). Thus, changes in rifting style, strain distribution, crustal architecture, timing, and nature of continental breakup can develop across distinct rifted margins and along the length of individual rift systems.

As most rift basins form along (or near) former orogens[6], inheritance is commonly invoked to explain the segmentation of both rifts and rifted margins[7,8]. Numerical modelling supports this inferred link between pre-rift structure and rifted margin architecture, with a wide range of 2-D investigations demonstrating the first-order effects of the initial thermal and rheological structure on continental rifting[9–14]. 2-D modelling also suggests that extension velocity[12–14], multiphase rifting[15,16], and complex deformation network localization[17–19] exert a first-order control on rifted margin structure. Furthermore, 3-D numerical simulations can now achieve similar spatial resolutions to 2-D studies and were used to illustrate the margin-parallel effects of structural inheritance[4,20–22], fault network coalescence[23] and out-of-plane[24,25] or oblique[26,27] boundary conditions. Nonetheless, to date no studies have explicitly examined the effects of a heterogeneous pre-rift lithosphere, with domains of varying rheology (i.e., composition, thickness, and thermal structure), on the 3-D evolution of continental rifting and rifted margin segmentation. This in part reflects that many rifted margins initiate on complex pre-rift lithosphere[6], which may be difficult to accurately reconstruct without sufficient data to connect onshore and offshore domains[28]. In the case of the Labrador Sea, geological, and geophysical data indicate an offshore along-strike segmentation that is clearly defined by onshore variations of crustal and lithospheric thickness, composition, and thermal structure[5,29,30].

The Labrador Sea formed between E Canada and SW Greenland as a branch of the North Atlantic Ocean[31]. The Early to Late Cretaceous rifting initiated on an Archean to Proterozoic cratonic lithosphere with different tectonic terrains amalgamated during the late Mesoproterozoic Grenvillian orogenic collision (ca. 1.08–0.97 Ga)[5]. Recent studies revealed important changes in rift geometry, crustal architecture, timing, and nature of breakup along the Labrador margin[5,32,33]. These changes occur across major Precambrian structures, which run perpendicular to the main NW rift trend and define three margin segments (Fig. 1). The southern segment has a Mesoproterozoic basement and shows a typical wide magma-poor rifted margin architecture with hyperextended continental crust and exhumed mantle. The central segment has a Palaeoproterozoic basement and also contains domains of hyperextended crust and exhumed mantle, but consists of volcanics in the continent-ocean transition not observed in the south. The northern segment, which formed on Archean terrains, displays a narrow magma-rich margin architecture with thick packages of seaward-dipping flood basalts and magmatic underplating (Fig. 1). Magnetic[34] and seismic[5] data indicate a diachronous continental breakup younging northward, with ca. 8 Myr gap between the onset of oceanic accretion in the southern segment and the northern segment (Supplementary Note 1). In

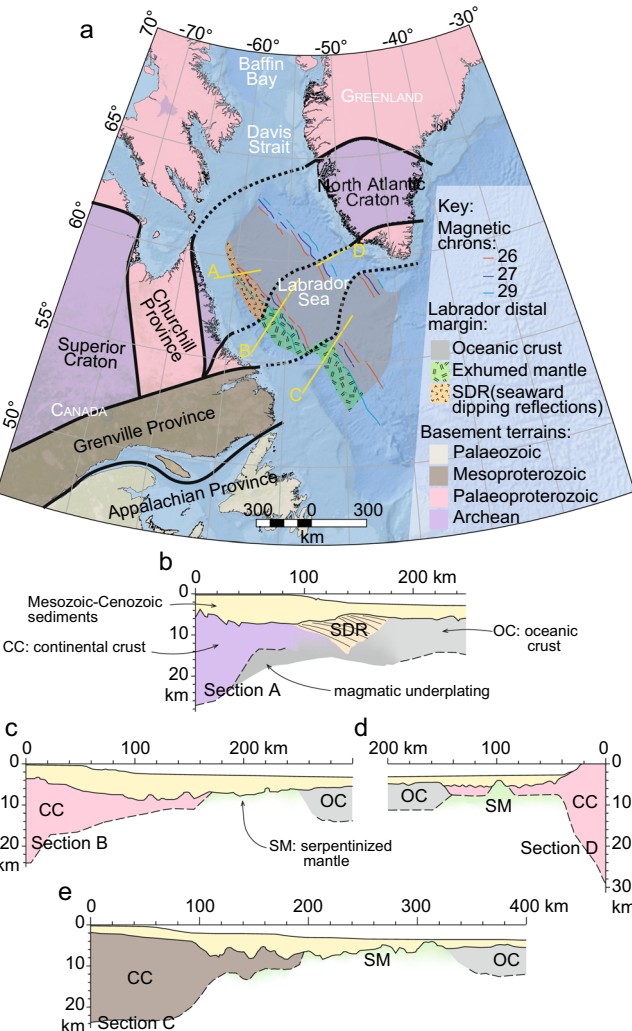

**Fig. 1 Geologic overview of the Labrador Sea and the surrounding un-rifted cratonic domains. (a)** Map of the Labrador Sea showing the rifted margins and the Archean-Proterozoic cratonic basement in onshore East Canada and South Greenland[5]. Cross-sections A–C illustrate the crustal architecture in the northern **(b)**, central **(c)**, and southern **(e)** segments of the Labrador margin[5], respectively. **(d)** Cross-section D shows the crustal architecture in the conjugate SW Greenland margin[39]. SDR: seaward-dipping reflections, CC: continental crust, OC: oceanic crust, SM: serpentinized mantle.

the latter, breakup appears to coincide with flood basalts located near the Davis Strait (ca. 61–56 Ma), whose link to the Iceland plume is suggested but still debated[35–37]. The crustal architecture, depicted by seismic data and gravity modelling, suggest an asymmetric rifting in the Labrador Sea (Fig. 1) and that the line of breakup was closer to the Greenland side (i.e., upper plate) than the Labrador side (i.e., lower plate)[5,32,33,38,39]. It implies that most of the crustal stretching is preserved on the Canadian side of the basin.

Geophysical data collected from the hinterland of the Labrador Sea (i.e., Canadian Shield) reveal the existence of variable heterogeneity in the un-rifted lithosphere of the distinct margin segments[29,30,40,41]. Seismic refraction across the Grenville suture[41] shows a crust, which is ca. 35 km thick in the Makkovik domain and ca. 50 km thick in the Grenville domain. A thickening attributed to underplating of a high velocity/density mafic crust during the Neoproterozoic-Early Cambrian Iapetan rifting[41]. In contrast, seismic tomography studies examining the

lithosphere structure in the Canadian Shield[29,30] reveal that the lithosphere-asthenosphere boundary is 200–150 km deep underneath the Grenville domain, but deepens to 200–250 km north of the Grenville suture. The observed lithosphere structure in the Labrador Sea and its hinterland is consistent with surface heat flow data[42], which show an E-W trend within the Labrador Sea, parallel to the rifted margin, and a N-S trend in the hinterland (i.e., Canadian Shield), parallel to the pre-rift tectonic domains[42]. The E-W trend is related to the Mesozoic rifting as heat flow steadily decreases from the distal margin to its hinterland. Whereas, the N-S trend is ascribed to pre-rift variation in both basal heat (i.e., lithosphere thickness) and crustal radiogenic heat production[42]. This suggests that the effect of the Labrador rifting on the thermal structure was restricted to the rifted domain and negligible in the margin hinterland.

Here, we use recent observations from the Labrador rifted margin[5] and thermo-mechanical modelling to examine the effects of variations in pre-rift lithosphere rheology on rift evolution and margin architecture. The numerical design and assessment are directly informed by the geological and geophysical constraints on pre-rift and syn-rift evolution, which are outlined above (see also Supplementary Notes 1–3). Our investigation reveals that inherited variations in lithosphere thickness, thermal structure and composition can reproduce key first-order observations from the Labrador Sea and should be considered when examining segmentation in rifts and rifted margins.

## Results

To examine the effects of these observed variations of lithospheric thickness, composition, and thermal structure on rift segmentation and margin architecture, we have developed thermo-mechanical models that assimilate the unique onshore geophysical constraints for each domain (Supplementary Fig. 1). While the observations sufficiently constrain first-order variations of crustal and lithospheric thickness (Supplementary Note 2), the rheology of distinct crustal layers is largely unknown. Consequently, a robust sensitivity analysis is necessary to account for the uncertainties in both crustal rheological layering and rates of extension. Given the extensive computational requirements of high-resolution 3-D simulations, our analysis uses a combination of 2-D and 3-D models, with the 2-D models constructed to represent the southern, central, or northern margin segments. We conduct the numerical experiments using the open-source finite element code ASPECT (v2.1.0-pre)[43,44], which is capable of efficiently solving for highly non-linear 3-D lithospheric deformation. The full results of the sensitivity analysis (Supplementary Movie 1–12) and model parameters (Supplementary Table 2) are presented in the Supplementary Information.

**Cratonic structure controls the timing and style of breakup.** The interplay between rheology and extension rate in each margin segment is assessed by running 2-D numerical experiments of slow (5 mm yr⁻¹, Fig. 2) and fast (10 mm yr⁻¹, Fig. 3) lithospheric extension with either an entirely weak crust (wet quartzite[45], Fig. 2a–b, e–f, Fig. 3a–b, e–f) or a weak upper crust and a strong (wet anorthite[46]) lower crust (Figs. 2c–d, g–h, Fig. 3c–d, g–h). Here, we focus on the end-member northern and southern segments. In all models, the lithosphere undergoes a three-stage rift evolution with a short-lived distributed stretching phase, followed by a localized necking phase, and finally a breakup phase (Supplementary Movie 1–12).

When the lower crust is weak (Supplementary Movie 1–3 and 7–9), the initial lithospheric stretching duration is governed by a variable crustal ductile layer thickness(es), which increases southward as the geothermal gradient and crustal thickness

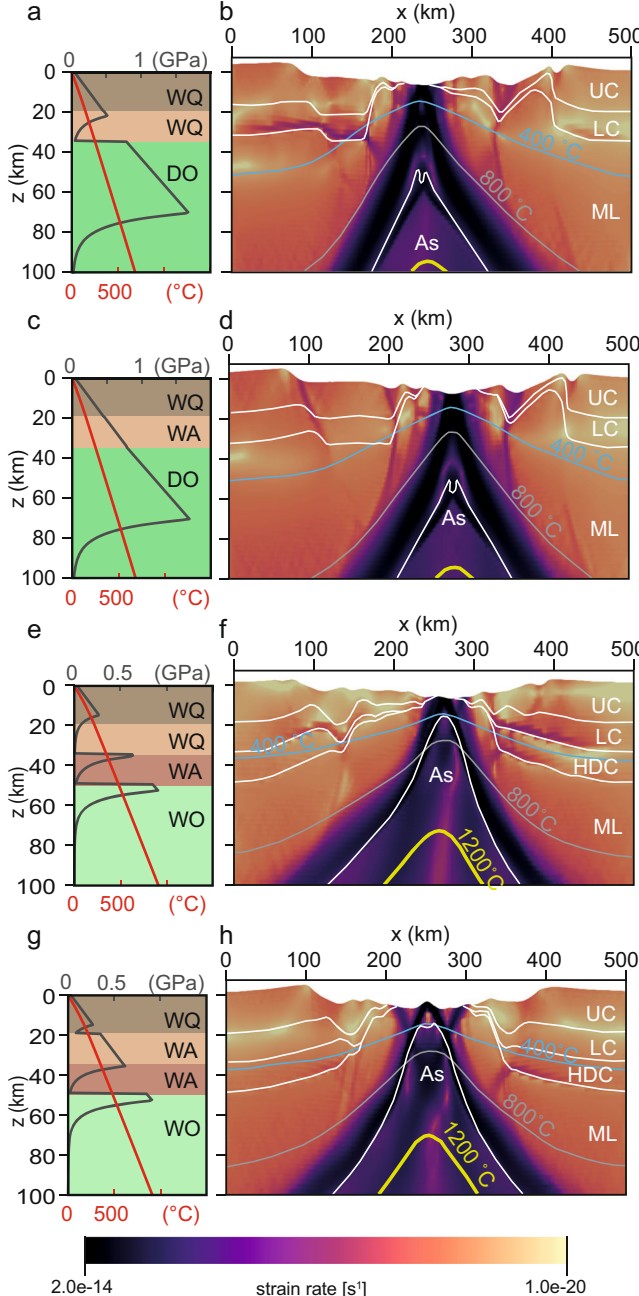

**Fig. 2 2-D models at 5 mm yr⁻¹ extension rate.** Lithospheric geometry at 40 Myr after rift initiation, assuming an extension rate of 5 mm yr⁻¹ for the northern segment (**a–d**) and the southern segment (**e–h**). Models with a weak lower crust (**a, b, e, f**) results in a wide and asymmetric crust geometry at crustal breakup, with hyperextended continental domain. Models with a strong lower crust (**c, d, g, h**) show a narrower and more symmetrical crust geometry at breakup, with no hyperextension. Shown are strength profiles (in dark grey), geotherms (in red), strain rate (in magma colourmap), isotherms at 400 (blue line), 800 (light grey line), 1200 °C (yellow line), and material boundaries (white lines; UC: upper crust, LC: lower crust, HDC: underplated high density crust, ML: mantle lithosphere, As: asthenosphere). WQ: wet quartz, WA: wet anorthite, WO: wet olivine, DO: dry olivine. Full simulation results are shown in Supplementary Movie 1–6.

increase (Fig. 2a–b, e–f, Fig. 3a–b, e–f). Once ductile flow in the upper lithosphere becomes negligible and coupling occurs between the brittle lithospheric layers, deformation localizes along a lithospheric-scale shear zone and ensuing rapid thinning

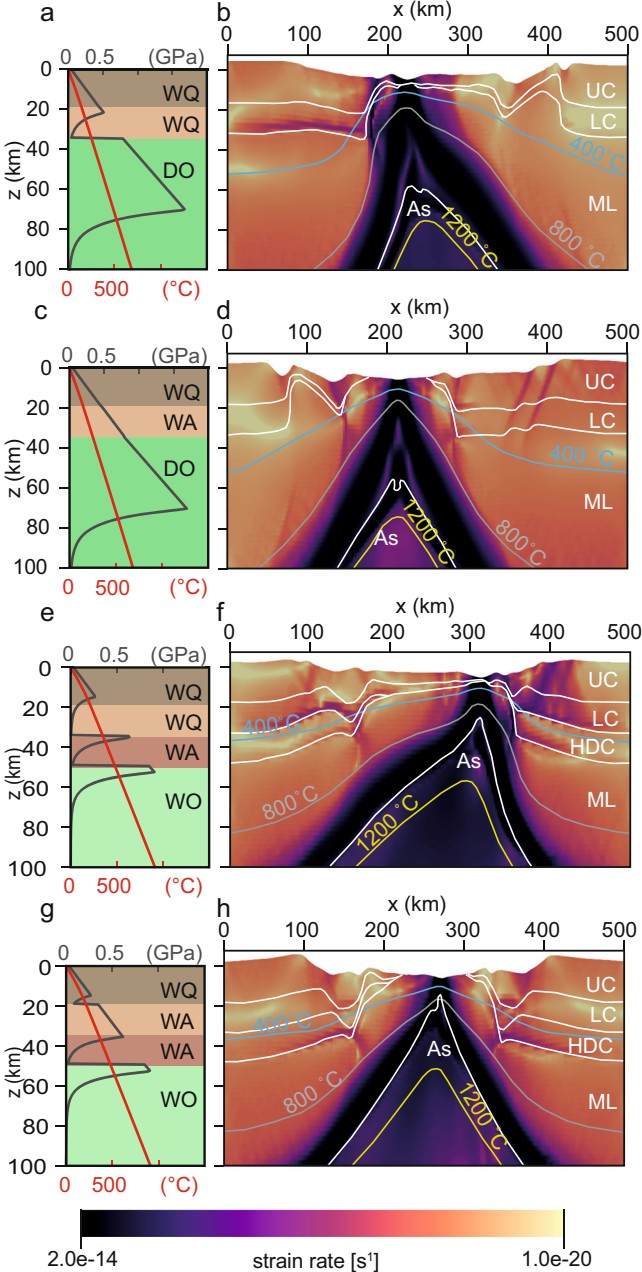

**Fig. 3 2-D models at 10 mm yr$^{-1}$ extension rate.** Lithospheric geometry at 20 Myr after rift initiation, assuming an extension rate of 10 mm yr$^{-1}$ for the northern segment (**a–d**) and the southern segment (**e–h**). The model assuming a weak lower crust in the northern segment (**a, b**) shows an asymmetric crustal geometry with a wide hyperextended domain, while the other models (**c–h**) show a narrower and symmetric crust geometry regardless of the rheology of the lower crust. Shown are strength profiles (in dark grey), geotherms (in red), strain rate (in magma colourmap), isotherms at 400 (blue line), 800 (light grey line), 1200 °C (yellow line), and material boundaries (white lines; UC: upper crust, LC: lower crust, HDC: underplated high density crust, ML: mantle lithosphere, As: asthenosphere). WQ: wet quartz, WA: wet anorthite, WO: wet olivine, DO: dry olivine. Full simulation results are shown in Supplementary Movie 7–12.

leads to the necking of the lithosphere. Early during the necking stage, extension switches from a pure to simple shear mode and results in the asymmetric hyperextension of the crust, crustal breakup, mantle exhumation, and finally full lithospheric breakup. Consistent with previous studies[12–14], an increase in

extension rate, a higher geothermal gradient, and a thicker crust promote a longer phase of hyperextension and thus a wider zone of hyperextended crust in the final rift configuration.

In the case of a strong lower crust (Fig. 2c–d, g–h, Fig. 3c–d, g–h, Supplementary Movie 4–6 and 10–12), ductile layers within the crust are absent and deformation initiates in a coupled manner between the crust and mantle lithosphere. In both the north and the south, the stronger crust suppresses the hyperextension phase and crust breakup occurs both earlier (less total strain) and in a symmetric manner. Prior to the full lithospheric breakup, the sub-continental mantle is exposed, and simple shear appears to dominate the final thinning of the mantle lithosphere. In contrast to the weak crust scenario, the extension rate, geothermal gradient and crustal thickness have significantly less impact on the final crustal architecture (Figs. 2 and 3).

These results support the hypothesis that observed crustal architecture variability along the Labrador margin is related to initial variation in geothermal gradient, thickness, and composition of the crust and the lithosphere. Furthermore, the results suggest that the initial lower crust was stronger in the north than in the south, which in combination with variations in the geothermal gradient enabled north-south gradients in the degree of hyperextension and margin asymmetry. However, the lack of margin-parallel deformation in the 2-D simulations could also neglect a process that produces margin segmentation without invoking a north-south change in crustal rheology. To test this finding, we conducted a suite of 3-D simulations (Fig. 4 and Supplementary Movie 13–18) that encompass the physical parameters examined with the 2-D simulations.

**Lateral rheological heterogeneities promote margin segmentation.** Our 3-D model design (Supplementary Fig. 1) captures the lateral rheological variability that is observed in the Canadian Shield, which is defined by variations in the geothermal gradient, crustal and lithospheric architecture. First, we conducted four 3-D simulations of a uniform lower crust composition, with either a strong or a weak lower crust rheology and an extension rate of 5 or 10 mm yr$^{-1}$ (Supplementary Movie 13–16). These scenarios do not produce any along-strike changes if the lower crust is strong, which results in a narrow rift system, a synchronous breakup, and a symmetrical conjugate margin system across all three segments (Supplementary Movie 14 and 16). Assigning a weak lower crust to the lithosphere induces some hyperextension in the southern segment (Supplementary Movie 13 and 15), where lithospheric breakup takes place 4 Myr earlier than in the other segments (in the case of 5 mm yr$^{-1}$ extension rate; Supplementary Movie 13).

However, imposing a variable lower crust composition with a strong lower crust in the northern segment and a weak lower crust in the central and southern segments leads to significant margin segmentation regardless of the extension velocity (Fig. 4 and Supplementary Movie 17 and 18). The temporal evolution (Fig. 4a–b) reveals that structural segmentation occurs early on in the stretching process as primary shear zones develop across the entire model crosscutting the rheological boundaries. Concurrently, secondary shear zones remain restricted to certain segments and either terminate or coalesce near segment boundaries (Fig. 4). The segmentation between the southern, central, and northern segments becomes more pronounced once the necking phase initiates. In the northern segment, where the lithosphere is cold and strong, deformation is coupled at the start of rifting and crustal necking leads to the complete thinning of the crust. Whereas in the southern segment, where the lithosphere is hotter and weaker and the crust is thicker, coupling is delayed and crustal necking leads to a prolonged phase of hyperextension. As a result, crustal breakup starts in the north

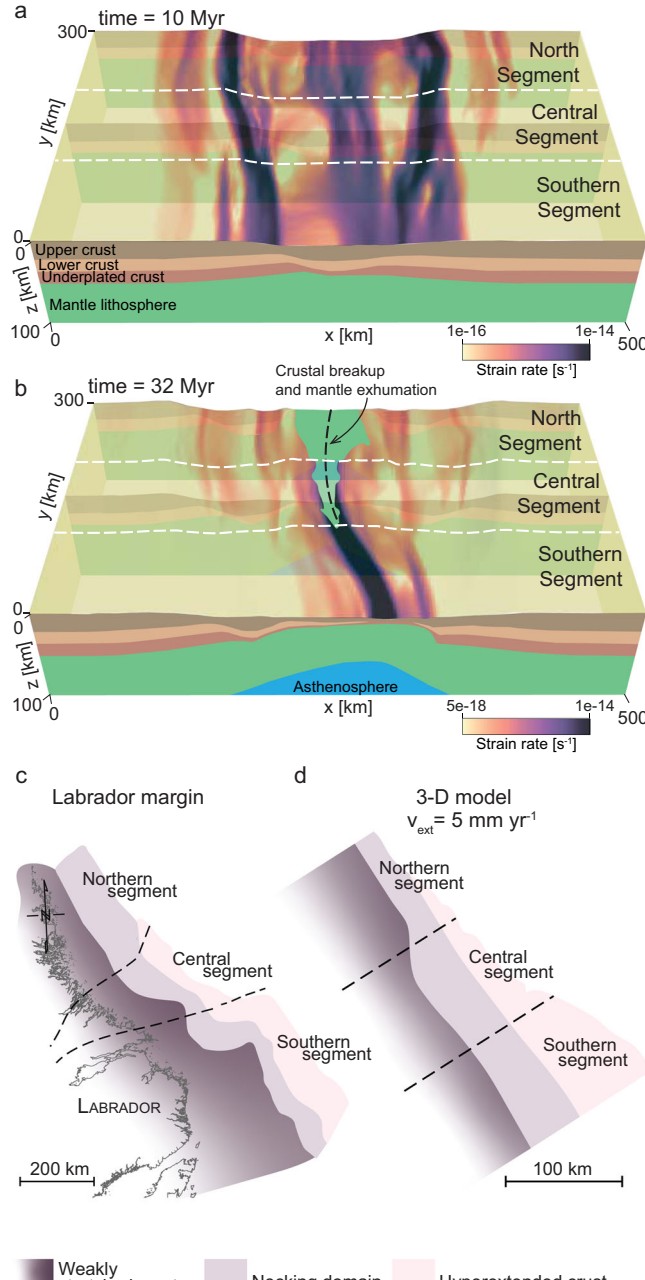

**Fig. 4 3-D modelling results.** The 3-D numerical experiment, at a slow extension rate ($v_{ext} = 5\,\mathrm{mm\,yr^{-1}}$), showing the distribution of active deformation (strain rate second invariant, magma colourmap), during the early stretching phase (**a**) and the late breakup stage (**b**). Structural segmentation is evident early on in the rifling process, with shear zones ending or coalescing at segment boundaries (**a**). Crustal segmentation occurs shortly after the necking stage when the crust breaks in the north while hyperextension is taking place in the south (**b**). Once full crustal breakup is accomplished along the model, the obtained crustal architecture (**d**) appears segmented with a narrow margin in the north (no hyperextension) and a wide hyperextended crust in the south. This is comparable to the crustal architecture observed along the Labrador margin (**c**). Full simulation results are shown in Supplementary Movie 17 and 18.

and propagates southward. In contrast, lithospheric breakup initiates in the south, where the initial mantle lithosphere is thinner, and propagates northward. The breakup process spreads over 18 Myr for the crust and 6 Myr for the mantle lithosphere.

The margin architecture at breakup shows a pronounced crustal segmentation between the three tectonic domains, with a crustal geometry that is narrow in the north and widens southward (Fig. 4d–e).

The change in rifting mode, rift geometry, and timing of breakup produced by the 3-D models are consistent with observations from the Labrador Sea[5] (Fig. 4c–e), which is characterized by a southward increase in margin width and crustal hyperextension, and a northward younging of lithospheric breakup.

**Extension rate and mantle composition govern magmatic budget during breakup.** The diachronous continental breakup along the Labrador margin is complemented by a northward increase in magmatic budget, with a continent-ocean transition that is magma-rich (i.e., SDRs) in the north and magma-poor (i.e., exhumed serpentinized mantle) in the south. The magma supply in the northern segment of the Labrador Sea is contemporaneous with the nearby Davis Strait Palaeocene flood basalts. The latter have been linked to adiabatic decompression melting due simply to the tectonics of rifting in the Davis Strait[36,37] or to excess temperature from the proto-Iceland plume[35]. In order to assess the potential of our modelling initial conditions in producing decompression melting during lithospheric stretching, we use the outputs from our two variable rheology 3-D models (i.e., with slow and fast extension) to compute melting. We extract the temperature and pressure fields at each time step of the models to calculate melt fraction based on the parametrization for batch melting of anhydrous and hydrous peridotite[47] (see parameters in Supplementary Table 3). Although this approach neglects the implications of melt-induced viscosity decrease and depletion of residual mantle on the geodynamics of rifting and melt production, it allows us to examine the likelihood of melt generation and its timing.

In the case of a dry depleted mantle (i.e., 0 wt% water), the thermal regime in slow rifting (i.e., $5\,\mathrm{mm\,yr^{-1}}$) remains colder than the anhydrous peridotite solidus[47], and fails to produce any melt (Fig. 5a), whereas, the fast rifting mode (i.e., $10\,\mathrm{mm\,yr^{-1}}$) is able to generate small volumes of melt but very late during the rifting process (Fig. 5a). In the second scenario, which assumes a hydrated enriched mantle (i.e., 0.05 wt% water), much larger melt volumes are produced and melt generation starts in the early rift stage regardless of the extension rate (Fig. 5a–d). In all cases, melt generation takes place first in the southern segment, where the initial geothermal gradient is higher, before reaching the central and northern segments, respectively, with up to 4 Myr delay (Fig. 5b).

These results show that the interplay between extension rate and mantle geochemistry could be the main factor controlling the magmatic budget during rifting and continental breakup in the Labrador Sea. A northward increase in extension rate and/or in mantle water (i.e., volatiles) content suffice to explain the increase in magmatism along the Labrador margin. Although the northward decrease in margin width indicates a decrease in the amount of extension, it does not necessarily correlate with a decrease in extension rate (see Supplementary Note 1 and Table 1). Lower to Upper Cretaceous syn-rift packages are well imaged in seismic reflection from the southern and central segments, providing a good constraint on the duration of rifting, whereas the lack of obvious syn-rift structures in the northern segment hinder any inference about the timing of extension[5,32,33]. On the other hand, tomography data from the Canadian Shield suggests that the Archean terrains and the surrounding Proterozoic to Palaeozoic domains are underlain by a geochemically different upper mantle[29,30]. The complex tectonics history of these domains

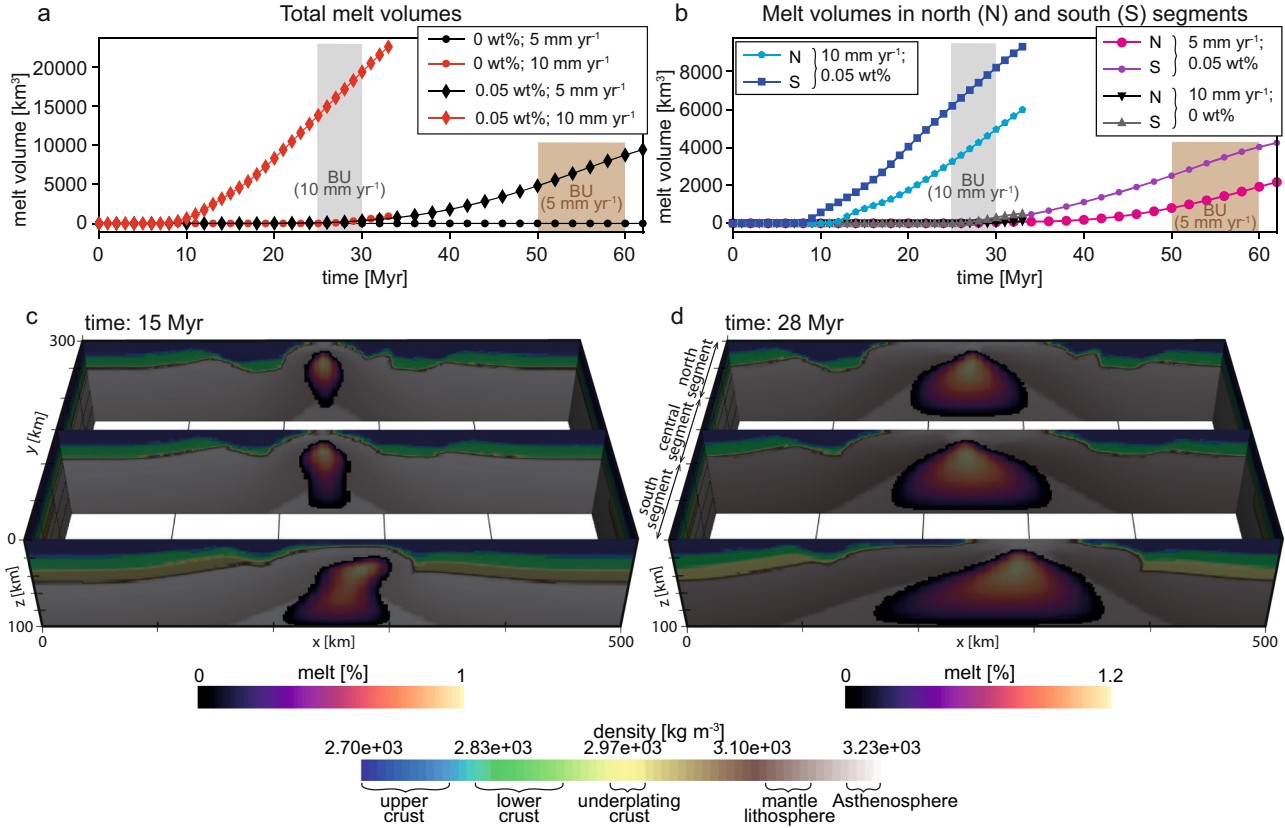

**Fig. 5 Decompression melt calculation.** Decompression melting generated during lithospheric stretching, using temperature and pressure outputs from the 3-D models (i.e., 5 and 10 mm yr$^{-1}$ extension rate). Calculations are based on the parametrization for batch melting of anhydrous and hydrous peridotite[47] (see parameters in Supplementary Table 3). We compute the volume of generated melt assuming different extension rates (5 and 10 mm yr$^{-1}$) and different water content in the mantle (0 and 0.05 wt%). The total melt volume produced at each time step is shown for the entire 3-D model (**a**) and for the northern (200 < Y < 300 km) and southern segment (0 < Y < 100 km) separately (**b**). Panels **c**, **d** show the melt generated at 15 and 28 Myr, respectively, assuming an extension rate of 10 mm yr$^{-1}$ and a mantle with 0.05 wt% water content. BU: continental breakup.

seems to result in variable degrees of chemical depletion of the underlying mantle lithosphere[30].

## Discussion

Our geodynamic models emphasise the role of pre-rift rheological heterogeneities in defining rift evolution and crustal architecture along rifted margins. The results of the 2-D experiments are consistent with previous numerical studies on the role of rheology in lithospheric [10–14], and demonstrate that the initial strength of the lower crust (i.e., composition and temperature), geothermal gradient, and the competition between frictional and viscous strain (i.e., decoupling) dictate the nature and timing of tectonic processes controlling lithospheric thinning. However, the 3-D models with variable lithosphere rheology between the southern, central, and northern segments provide new insights into how lateral rheological variation can promote segmentation along rifted margins.

The segmentation is not only expressed by a reactivation of pre-existing structures (e.g. sutures and shear zones), as shown in previous studies, but can also be driven by a change in the processes controlling rifting within each segment, as indicated by our 3-D models. It manifests laterally in the distribution of rift structures, the variability in crustal architecture, and the change in the timing and magmatic budget of continental breakup. In addition, this magmatic budget variability is more likely to be driven by changes in the extension rate during rifting and/or changes in the geochemistry (i.e., fertility/depletion) of the mantle underneath the rifted plate.

We note that most Atlantic margins exhibit some degree of segmentation and along-strike variability[3]. The Scotian margin,

for instance, shows major changes in the crustal architecture and the magmatic budget during breakup, with a magma-poor segment in the north and a magma-rich segment in the south[48]. Whereas the Norwegian margin, on the other hand, appears to be affected by a strong structural segmentation from the earliest phases of extension preserved in the proximal domain[49,50] to segmentation along the necking and hyperextension-exhumation domains[51]. Unlike the Labrador Sea, which opened perpendicular to Precambrian terrains, these margins initiated on, and parallel to, Palaeozoic Appalachian and Caledonian terrains, respectively, with a substantially more complex structural inheritance.

Indeed, recent detailed mapping along the proximal Norwegian margin demonstrates strong structural inheritance at a range of scales[49]. Nonetheless, because the basement terrains do not continue into the un-rifted onshore domain, their pre-rift lithospheric properties (i.e., composition and thermal structure) remain enigmatic. While the assimilation of such pre-rift structural inheritance into numerical simulations presents significantly more challenges than the simulations presented here, they also provide new opportunities to rigorously examine the relationship between inherited shear zone rheology and extensional reactivation. In examining such complex structural inheritance scenarios, future modelling efforts may include the effects of grain size evolution and strain healing to help provide further constraints on lithospheric and rifted margin processes.

The Labrador Sea provides an ideal natural laboratory to constrain the pre-rift variations of the basement lithospheric properties and examine their impact on rifting processes, segmentation, and

nature of continental breakup. Our findings show that variations in heat flow, lithospheric and crustal thickness and composition exert a first-order control on the evolution of rifted margins, which in most cases initiate on a lithosphere with strong inherited lateral heterogeneities. We also demonstrate that observations from margin hinterland could be used as a proxy to gain insights into the initial (thermal and compositional) state of the lithosphere.

## Methods

**Governing equations**. We model the thermal-mechanical evolution of continental extension in a heterogenous initial lithosphere using the open-source and CIG-supported finite element code ASPECT version 2.0.1-pre[43,44]. The ASPECT version and parameter files required to reproduce our experiments can be found in the following GitHub branch: https://github.com/naliboff/aspect/tree/labrador_sea_gouiza_naliboff_2020.

Velocity and pressure are solved for assuming incompressible viscous flow, where the Stokes equations are defined as:

$$\nabla \cdot u = 0 \tag{1}$$

$$-\nabla \cdot 2\mu\dot{\varepsilon}(\mathbf{u}) + \nabla p = \rho\mathbf{g} \tag{2}$$

Above, $\mathbf{u}$ is velocity, $\mu$ is viscosity, $\dot{\varepsilon}$ is the strain rate, $p$ is pressure $\rho$ is density, and $\mathbf{g}$ is gravity.

Temperature evolves through a combination of advection, heat conduction, shear heating, and adiabatic heating:

$$\rho C_p \left(\frac{\partial T}{\partial t} + \mathbf{u} \cdot \nabla T\right) - \nabla \cdot (\kappa\rho C_p)\nabla T = \rho H + 2\eta\dot{\varepsilon}(\mathbf{u}) + \alpha T(\mathbf{u} \cdot \nabla p) \tag{3}$$

where $C_p$ is the heat capacity, $T$ is temperature, $t$ is time, $\kappa$ is thermal diffusivity, and $H$ is the rate of internal heating. Respectively, the terms on the right-hand side correspond to internal head production, shear heating, and adiabatic heating.

Density varies linearly as a function of the reference density ($\rho_0$), thermal expansivity ($\alpha$), reference temperature ($T_0$), and temperature:

$$\rho = \rho_0(1 - \alpha(T - T_0)) \tag{4}$$

**Rheological formulation**. The constitutive behaviour combines non-linear viscous flow with brittle failure[52], with viscous flow following a dislocation creep (Eq. (5)) in the lithosphere:

$$\sigma'_{II} = A^{-\frac{1}{n}}\dot{\varepsilon}_{II}^{\frac{1}{n}}e^{\frac{Q+PV}{nRT}} \tag{5}$$

Within the asthenosphere, viscous flow is a composite (harmonic average) of dislocation and diffusion creep (Eq. (6)):

$$\sigma'_{II} = A^{-1}\dot{\varepsilon}d^{\frac{p}{n}}e^{\frac{Q+PV}{nRT}} \tag{6}$$

Above, $\sigma'_{II}$ is the second invariant of the deviatoric stress, $A$ is the viscous prefactor, $n$ is the stress exponent, $\dot{\varepsilon}_{II}$ is the second invariant of the deviatoric strain rate (effective strain rate), $Q$ is the activation energy, $P$ is pressure, $V$ is the activation volume, $T$ is temperature, and $R$ is the gas constant, $d$ is grain size, and p is the grain size exponent.

Brittle (plastic) behaviour follows a Drucker-Prager yield criterion formulation, where the yield stress in 3-D is a function of the cohesion ($C$), angle of internal friction ($\phi$), and pressure ($P$):

$$\sigma'_{II} = \frac{6C\cos\phi + 2P\sin\phi}{\sqrt{(3)}(3 + \sin\phi)} \tag{7}$$

To help localize deformation and account for geologic observations of strain localization, we track the accumulation of plastic strain (invariant form) and weaken the cohesion and friction linearly by a factor of 4 between plastic strain values of 0.5 and 1.5[15,19].

The procedure for calculating the viscosity at every point follows the viscosity rescaling method, which first compares the predicted effective stresses from viscous flow and plastic failure. If the viscous stress exceeds the plastic yield stress, the viscosity is reduced so the effective stress exactly matches the plastic yield stress[23,52].

**Discretization and non-linear solvers**. Throughout the model domain we use quadratic elements (Q2) elements to solve the advection-diffusion equation for temperature, while the Stokes equation is solved on elements that are quadratic for velocity and continuous linear for pressure (Q2Q1). The element size is 10 km beneath 300 km depth, 5 km from 100 to 200 km depth, and 2.5 km above 100 km depth. In total, the 3-D numerical simulations contain ~463 million degrees of freedom. Compositional fields are used to track and advect distinct lithologic domains (e.g. rock types) and other time-dependent quantities (strain). The use of discontinuous Galerkin element with a limiter for compositional fields[53] minimizes diffusion of distinct layers and improves the accuracy of interface advection through time.

Nonlinearity introduced by the constitutive model is resolved using standard Picard iterations for the velocity and pressure to a tolerance of $10^{-4}$. In most models we use a conservative maximum time step of 20,000 years to limit numerical instabilities during advection and improve the non-linear convergence behaviour. This value is adjusted proportionally as the boundary velocity values increase or decrease. We use this outlined numerical approach to construct a series of 2-D and 3-D continental rifting simulations that reveal the relationship between initial lithospheric structure and rifted margin structure. In order to provide a robust sensitivity analysis of key modelling variables, we first carefully consider the geologic constraints on the pre-rift lithospheric structure.

**Initial conditions and material properties**. The simulations in this study use a wide range of initial conditions (composition, temperature) and material properties to characterize different scenarios for pre-rift lithospheric structure. The details of these simulation features and the methods used to define them are outlined in Supplementary note 3.

**Future model improvements**. The forward modelling software used in this study (ASPECT) is community driven, actively developed, and used for a wide range of mantle convection and lithospheric dynamics investigations. In the duration since the models in this study were designed and run, development on portions of the code relating to lithospheric rheology, adiabatic heating, material tracking, free surface stabilization, and non-linear solver schemes has occurred. While these changes do not affect the first- or second-order conclusions reached in this study, future investigations should take advantage of new or updated functionality. Readers wishing to build on the results presented here are encouraged to contact the authors for advice and guidance on how to modify the provided parameter files to take advantage of these changes.

## Data availability

The modelling results generated in this study have been deposited in the Mendeley Data Repository (https://doi.org/10.17632/9h3vjvn2ms.1).

## Code availability

The code ASEPCT, used in this work, is open source and can be downloaded at https://aspect.geodynamics.org/. The code version of ASPECT and parameter files required to reproduce our experiments are freely available on GitHub at https://github.com/naliboff/aspect/tree/labrador_sea_gouiza_naliboff_2020. We encourage any readers that would like to use the methodology and parameter files presented here to contact us in order to take advantage of recent improvements to ASPECT. ASPECT is built on the open-source finite element package deal. II, which we built (version 9.0.0) through the candi installation package (https://github.com/dealii/candi). Additional dependencies built through candi include Trilinos (12.10.1) and p4est (2.0.0). Detailed instructions for building ASPECT and deal. II on the XSEDE-supported supercomputer Stampede2 are available at https://github.com/geodynamics/aspect/wiki.

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

## Acknowledgements

We thank the Computational Infrastructure for Geodynamics (geodynamics.org), which is funded by the National Science Foundation under award EAR-0949446 and EAR-1550901 for supporting the development of ASPECT. The 2-D modelling was undertaken on ARC4, part of the High-Performance Computing facilities at the University of Leeds, UK. The computational time for the 3-D simulations was provided under XSEDE project EAR180001. The work described in this paper was primarily funded through the Basin Structure Group (BSG, University of Leeds) and by the Computational Infrastructure for Geodynamics (NSF award 1550901).

## Author contributions

M.G. conceived the original idea for the investigation, co-designed and tested the numerical simulations, analysed and visualized the simulations, and co-wrote the publication. J.N. co-designed and tested the numerical simulations and co-wrote the publication.

## Competing interests

The authors declare no competing interests.
