## [Peer Review File · Nature Communications]

REVIEWER COMMENTS

Reviewer #1 (Remarks to the Author):

Dear Authors,

I have now read your paper titled: "Rheological inheritance controls the formation of segmented rifted margins in cratonic lithosphere". I think that the paper is generally well written and on a very relevant topic. The combination of 2D and 3D models works well. The quality of the figures is also very high. Overall, I think that the paper should eventually be published following the revisions outlined below. My main concerns relate to the presentation of the work in the context of very similar previous studies and also the justification of the study area. In addition, I note a number of minor points that should also be considered.

Major points:

1) Previous work 3D modelling the NW Atlantic: Heron et al. (2019), Santimano & Pysklywec (2020), Clarke & Beutel (2019) as well as other numerical modelling studies.

The study by Heron et al. (2019) uses a very similar 3D numerical modelling approach to investigate very similar processes primarily focused on the Davis Strait region but also including the Labrador Sea and Baffin Bay (i.e. the whole NW Atlantic - as defined by Abdelmalak et al. 2018). Both the present study and Heron et al. (2019) use the ASPECT numerical model and both are investigating tectonic inheritance and margin segmentation. In addition, Santimano & Pysklywec (2020) also used 3D numerical models to investigate the NW Atlantic, albeit slightly to the north, but still relevant to the present study. Whilst Clarke & Beutel (2019) also conducted 3D numerical modelling experiments of the region looking at stress concentration and implications for magmatism. I therefore think it is not correct when the authors state that "to date no studies have explicitly examined the effects of variable lithospheric structure on the 3D evolution of continental rifting and rifted margin segmentation". Not only have such models been produced elsewhere (e.g. Koopmann et al., 2014), but similar models have actually already been produced for the present study area. However, I don't think this necessarily means that the present study is not sufficiently original because the work of Heron et al. (2019) is focused on the role of mantle structure whereas the present study is focused on shallower inheritance, and the other studies mentioned are focused on different aspects of rift evolution. I do think however that a much greater discussion and integration of the results with these studies, and in particular the work of Heron et al. (2019) is needed. Currently I feel it is a bit disingenuous of the authors to be claiming that this the first study of its kind in the region. To reiterate though, I don't think this is necessarily a reason for rejection, or a major flaw of the study, but it does mean that a substantial change to the manuscript is required.

2) Nature of segmentation on the Labrador margin and elsewhere in the region: A question of scale?

This comment has two parts a) and b).

a) Previous work on the north west Atlantic does indeed note significant rift segmentation (Heron et al., 2019). However, the major segmentation of this rift system is into 3 major regions. From south to north these are: 1) the Labrador Sea, 2) the Davis Strait and 3) Baffin Bay. Of course, it is possible that secondary and smaller scale segmentation of these regions (i.e. the Labrador Sea) exists but this is very much secondary, and it needs to be presented in the context of the larger scale rift segmentation which previous work also links to structural inheritance (e.g., Peace et al., 2018b).

b) Related to the point a), I have some comments on the actual nature of segmentation in the Labrador Sea that I feel need addressing also. Considering the crustal structure of the Labrador sea region as presented by Welford and Hall (2013) following their gravity inversion study, it is not entirely clear to me that significant segmentation of the margin occurs at the crustal scale as implied by the present study. In addition, Peace et al. (2017) present a LAB (lithosphere–asthenosphere boundary) depth model for the entire NW Atlantic (Fig. 5b in Peace et al. 2017). Again, the LAB depth model does not really show

significant segmentation on the Labrador margin. As such, I feel that the authors should present their evidence for segmentation of the Labrador margin in the context of this work. Is the evidence for the segmentation based around the different geological terrains that the Labrador Sea bisects? Essentially, I just felt that a better discussion of the nature of the segmentation based on the previous geological and geophysical work in the region was needed.

3) Nature of the SW Greenland margin and incorporation of this into models.

Currently, the present study by Gouiza and Naliboff focuses primarily on the Labrador margin. This is understandable given that much more is known about the Labrador margin than its SW Greenland conjugate. However, an abundance of previous work has concluded that to fully comprehend a rift prior to breakup both conjugates must be considered in models (e.g., Wilson et al., 2001). This is particularly relevant in the Labrador Sea where the structure of the margins has been shown to be asymmetric (Peace et al., 2016). The obvious implication of this asymmetry is the question of whether the apparent segmentation that forms the focus of the study by Gouiza and Naliboff also occurs in SW Greenland? Overall, I think that the variable structure of the two conjugates should be expanded upon more with particular emphasis on the segmentation aspect.

4) Previous work specifically on structural inheritance in the NW Atlantic

The present study by Gouiza and Naliboff is essentially modelling structural inheritance at a very large scale. Given this, I felt that further discussion of the work in the region looking at inheritance is needed (e.g., Wilson et al., 2006; Peace et al., 2017, 2018a, 2018b).

5) Choice of the study area

The authors imply that the Labrador Sea is one of the only segmented rifts globally (Line 39). However, this is clearly not the case as many very recent studies are focused on this phenomena (Heron et al., 2019; Scholz et al., 2020). I felt when reading the text that the authors should better justify the choice of study area.

6) Mantle Plumes?

Given that the models presented by Gouiza and Naliboff include variable thermal conditions as a starting point, I am wondering if any inferences can be made about the hypothesised mantle plume in the region (e.g., Gerlings et al., 2009)? This may not be applicable, but I am wondering if any of the features sometime attributed to a plume (e.g., regional breakup) occur in the models in the absence of a plume?

Minor points:

Title: Why doesn't the title mention the Labrador Sea when the whole paper is focused on this region?

Lines 11-12: I am not sure that margin segmentation is completely enigmatic, lots of previous work links it to inheritance.

Lines 13-14: I disagree with this statement and I would say that previous studies have made progress here. See for example Heron et al. (2019) which is literally modelling the same region as the present study.

Line 15: "thermal-mechanical modelling" – might be worth mentioning ASPECT here?

Line 15: "continental rifting" – And breakup?

Line 17: "Lithospheric thickness" - Peace et al. (2016) show a figure that implies there is minimal variation in LAB (lithosphere-asphenosphere Boundary) depth along the margins of the Labrador Sea. The major structure is associated with the Davis Strait to the North. In addition, the crustal thickness from gravity

inversion shown in Welford and Hall (2013) shows minimal variation along the Labrador margin.

Line 26: "segments" - So the major segmentation of the NW Atlantic region is the Labrador Sea - Davis Strait - Baffin Bay. I think it would help if the authors could justify why they are focusing on a part of this system with less segmentation than elsewhere in the NW Atlantic? (I realise that this is also the subject of one of the more major points above).

Lines 34-35: "Despite these advances, to date no studies have explicitly examined the effects of variable lithospheric structure on the 3D evolution of continental rifting and rifted margin segmentation" - I disagree with this statement.

Line 53: But what about the SW Greenland margin?

Line 61-62: It is not universally accepted that the volcanic rocks in the Davis Strait are plume related (Peace et al., 2017; Clarke and Beutel, 2019).

Line 66: Again, what about the SW Greenland margin? Can you comment on the structure of this margin?

Line 95: The asymmetric extension is explicitly studied by Peace et al. (2016)

Line 137-138: Ok, but so do lots of previous studies. I think it would be good to explicitly state what is different about this study.

Figure 1: This is a very nice figure and gives a good regional overview. My one comment would be that it really does emphasize the "1-sided" approach to studying the Labrador Sea rift and shows how the SW Greenland margin has been sort of ignored. There are openly available crustal scale seismic lines on the SW Greenland margin (Chian et al., 1995) that could be considered here.

Figure 2: This figure gives a very clear overview of the modelling setup.

Figure 3: I found this figure to be a bit hard to get into. I am sure there is a reason but why are the 8 Myr images shown on the left and the 4 Myr images on the right? I appreciate that a lot of different model iterations are being shown on one figure but I found this figure quite hard to interpret. Can it be separated out into more figures or is that not allowed in the journal?

Figure 4: This is a very nice figure and shows off the 3D model nicely.

Kind regards,

Dr. Alexander L. Peace
McMaster University, Canada

References cited in this review

- Abdelmalak, M.M., Planke, S., Polteau, S., Hartz, E.H., Faleide, J.I., Tegner, C., Jerram, D.A., Millett, J.M., and Myklebust, R., 2018, Breakup volcanism and plate tectonics in the NW Atlantic: *Tectonophysics*, v. 760, p. 267–296, doi:10.1016/j.tecto.2018.08.002.
- Chian, D., Loudon, K.E., and Reid, I., 1995, Crustal structure of the Labrador Sea conjugate margin and implications for the formation of nonvolcanic continental margins: *Journal of Geophysical Research*, v. 100, p. 24239, doi:10.1029/95JB02162.
- Clarke, D.B., and Beutel, E.K., 2019, Davis Strait Paleocene Picrites: Products of a Plume or Plates? *Earth-Science Reviews*, doi:10.1016/j.earscirev.2019.01.012.
- Gerlings, J., Funck, T., Jackson, H.R., Loudon, K.E., and Klingelhofer, F., 2009, Seismic evidence for plume-derived volcanism during formation of the continental margin in southern Davis Strait and northern Labrador Sea: *Geophysical Journal International*, v. 176, p. 980–994, doi:10.1111/j.1365-246X.2008.04021.x.

- Heron, P.J., Peace, A.L., McCaffrey, K.J.W., Welford, J.K., Wilson, R.W., van Hunen, J., and Pysklywec, R.N., 2019, Segmentation of rifts through structural inheritance: Creation of the Davis Strait: *Tectonics*, doi:10.1029/2019TC005578.
- Koopmann, H., Brune, S., Franke, D., and Breuer, S., 2014, Linking rift propagation barriers to excess magmatism at volcanic rifted margins: *Geology*, v. 42, p. 1071–1074, doi:10.1130/G36085.1.
- Peace, A.L., Dempsey, E.D., Schiffer, C., Welford, J.K., Ken, J.W., McCaffrey, K., Imber, J., and Phethean, J.J.J., 2018a, Evidence for Basement Reactivation during the Opening of the Labrador Sea from the Makkovik Province, Labrador, Canada: Insights from Field Data and Numerical Models: *Geosciences*, v. 8, p. 308, doi:10.3390/geosciences8080308.
- Peace, A.L., Foulger, G.R., Schiffer, C., and McCaffrey, K.J.W., 2017, Evolution of Labrador Sea–Baffin Bay: Plate or Plume Processes? *Geoscience Canada*, v. 44, doi:10.12789/geocanj.2017.44.120.
- Peace, A., McCaffrey, K.J.W., Imber, J., van Hunen, J., Hobbs, R., and Wilson, R., 2018b, The role of pre-existing structures during rifting, continental breakup and transform system development, offshore West Greenland: *Basin Research*, v. 30, p. 373–394, doi:10.1111/bre.12257.
- Peace, A.L., McCaffrey, K.J.W., Imber, J., Phethean, J., Nowell, G., Gerdes, K., and Dempsey, E., 2016, An evaluation of Mesozoic rift-related magmatism on the margins of the Labrador Sea: Implications for rifting and passive margin asymmetry: *Geosphere*, v. 12, doi:10.1130/GES01341.1.
- Santimano, T., and Pysklywec, R., 2020, The influence of lithospheric mantle scars and rheology on intraplate deformation and orogenesis: Insights from tectonic analogue models: *Tectonics*, p. 1–19, doi:10.1029/2019tc005841.
- Scholz, C., Shillington, D., Wright, L., Accardo, N., Gaherty, J., and Chindandali, P., 2020, Intrarift fault fabric, segmentation, and basin evolution of the Lake Malawi (Nyasa) Rift, East Africa: *Geosphere*, v. 16, p. 1–19, doi:10.1130/GES02228.1.
- Welford, J.K., and Hall, J., 2013, Lithospheric structure of the Labrador Sea from constrained 3-D gravity inversion: *Geophysical Journal International*, v. 195, p. 767–784, doi:10.1093/gji/ggt296.
- Wilson, R.W., Klint, K.E.S., van Gool, J.A.M., McCaffrey, K.J.W., Holdsworth, R.E., and Chalmers, J.A., 2006, Faults and fractures in central West Greenland: onshore expression of continental break-up and sea-floor spreading in the Labrador–Baffin Bay Sea: *Geological Survey Of Denmark And Greenland Bulletin*, v. 11, p. 185–204, http://www.dur.ac.uk/k.j.w.mccaffrey/McCaffrey_pdfs/Wilsonetla_GEUS07.pdf.
- Wilson, R.C.L., Whitmarsh, R.B., Taylor, B., and Frotzheim, N., 2001, Introduction: the land and sea approach: *Geological Society, London, Special Publications*, v. 187, p. 1–8, doi:10.1144/GSL.SP.2001.187.01.01.

Reviewer #2 (Remarks to the Author):

Dear authors,

I have read with attention the paper entitled “rheological inheritance controls the formation of segmented rifted margins in cratonic lithosphere” by Gouiza and Naliboff and I think the idea developed in the paper is new and good, the sciences is despite some typo in the parameters table of good quality, the literature cited is adequate and figure and supplementary movies are informative.

The paper reads well but I really think it would definitely benefit of being lengthened to better include the geological arguments that are mostly given in the supplementary material.

As in all modeling study that are related to inheritance, the model set up and all the heterogeneities included in the model will for sure control the results that is no surprise so that means that extra care must be given to argument and justify all these heterogeneities.

In that paper main variations are caused by

1/ Change in lithospheric thicknesses that are not sufficiently argued (why these 3 values, are you taking todays thickness ? from which tomographic models ? why should it be the same 100 Myr ago ?)

2/ Change in upper crust heat production (there are no arguments but this is extremely important effects on the geotherm).

3/ Justification for changes in crustal thickness is not clear either, at least it is not straight forward in figure 1 and it is not discussed in the supplementary material. After reading all the material I still wonder what is the underplated material in the grenville belt.

4/ Also the supplementary material mentions that the lithosphere of the grenville is more depleted but that does not show on figure 2 and it is not mentioned in the main text. Are you using a different mantle lithosphere type on the Southern profile? If so please put a different coloring on figure 2 and clarify which creep law from supplement table is used in the caption of figure 2.

I am not saying the parameters chosen are not correct, I just want to emphasize that they are not enough justified in the text.

Also the paper would benefit from a bit more discussion on propagation of crustal and lithospheric break up in the 3D models and how this fits with geological observations and noteworthy enough with the presence and absence of magmatism... The models do not include any Iceland plume, does it means we don't need a plume to explain the opening of the Labrador Sea and the SDR?

So think this paper should be published and that if the authors were to add more geological and geophysical arguments for their set up, and a more details discussion on the significance of their results on rift propagation (comparing with other 3D modelling studies) and on the plume / SDR issue in the main text , it will significantly improve its impact.

Below I least minor comments and typo that need to be fixed but did not arm my understanding of the paper because these are clearly typos and I like to thanks the author for this easy to read and informative paper.

Laetitia Le Pourhiet

Minor comments

1/ Your last concluding statement l. 150 should be removed, it is not true.

You have assumed that it is true, but you have not demonstrated it is true with your contribution.

2/ equations l. 215 - 223

please add spaces after variables, check the accents on strain rate should be a dot... make u and g bold faced since they are vectors. I know it is hard typing equations in ms word but really this is ugly and I would be ashamed to be thanked as a reviewer if you leave it like this.

Please use k not K for thermal conductivity and rather say that you conserve heat rather than advection diffusion equation... or you should write advection diffusion production of heat equation.

Also you don't need to write deviatoric strain rate since you assume that divergence of velocity field is zero, it is just the strain rate.

Do not put brackets around friction angle in the yield criteria but make sure \sin and \cos are not italic but regular these are functions not variables.

Maybe use II subscript rather than eff since it is the second invariant, first being the pressure.

You mention diffusion creep in your table of parameters but there is no description on how you account for it in the description of the rheology.

3/ Supplementary table:

Why do you have 50 kg/m^3 difference in density between the asthenosphere and lithosphere at reference temperature? I understand that you consider that it is depleted in some case but not depleted in the others...

but if it is not depleted there is no reason for the lithosphere to be less dense at similar temperature if the Mg number is the same

Please add units for the prefactor A should be $\text{Pa}^{-n} \text{s}^{-1}$

You give heat equation with conductivity and here you list diffusivity. Either you use constant diffusivity or constant conductivity in the code (I don't know actually) but since the density depends on temperature would be good to write the heat conservation equation according to the quantity you consider as constant.

Thermal expansivity must be 2.5×10^{-5} not $+5$

The table mention 0.25 for strain weakening factor but the text says it varies from 0.5 to 0.25 (actually you write a factor 2 or 4... would be nice if the table is more consistent with the text)

Please add references for the value used for heat production in the upper crust and justify them because the geotherm and changing heat production at the surface changes the temperature of the moho and the depth of the 1300°C isotherm, so it is very important in your study.

I believe the underplated crust must be 0.02×10^{-6} not 0.02×10^{-4} (otherwise we would see a bump in the geotherm)

4/ Figure 2 :

Please change the vertical scale in 2a to be compatible with 2b

Reviewer #3 (Remarks to the Author):

The paper "Rheological inheritance controls the formation of segmented rifted margins in cratonic lithosphere" by Gouiza and Naliboff is an interesting contribution on the evolution of continental rifts into oceanic basins. This relatively-short phase has a relevant control on the formation of a fundamental feature of our planet, the ocean floor. In fact, the complexities of the rifting stage remain inherited in oceanic lithosphere and control the geometry and growth of these basins for very long time. To date, our understanding of this stage is still poor, this is mostly due to the simplification to a two-dimensional problem, originally invoked for conceptual necessity and computational limitation. However, this simplification has become an obvious limitation. This is now overcome, in the paper by Gouiza and Naliboff. Testing how rifts evolve in three-dimension around pre-existing continental heterogeneities is an elegant solution that clearly advance our understanding.

The modelling is neat and clear although the comparisons and presentation should be improved before the paper is published. The writing is a bit too generic in many places and the structure could be improved: If the geological evidence is presented upfront, then there must be a section where the model's outcomes are compared with the observables. Of course, if so, it is important to avoid repetition.

The figures do not serve well the paper. The figure 3 does not directly compare to the observation in figure 1, so it remains quite difficult to understand how well this suite of models reproduce the Labrador Sea case. Also, figure 4 is very interesting, however, this should be presented in more detail. For instance, cross section through the 3 segments could be shown. At this stage, the 2D models could be deemed redundant, I guess.

Most importantly, the comparisons between the top surface deformation and the geological case is critical, the hand-drawing on the model does not provide much information. Perhaps a strain rate field or something

else would be better.

I would recommend to give the figures a thought, it is a short paper and using one figure for the model set up and one for cramming all the 2d models might not be a helpful strategy: while it is perfectly fine for a long paper, in such a letter this reduces the audience and the impact.

Melbourne 25 September 2020

Fabio A. Capitanio

I provide below some more specific comments

Abstract

The opening sentence "Observations from rifted margins reveal that significant structural and crustal heterogeneity develops through the process of continental extension and breakup" is not very clear and misleading: it emphasises the heterogeneities during the rifting, whereas the paper focuses on those pre-existing.

Line 14 and ff. "Here, we use recent observations from the Labrador Sea and thermal-mechanical simulations of continental rifting to constrain the effects of inherited variable lithospheric properties on margin segmentation. The modelling results demonstrate ..." This is not a recommended way to proceed. In fact, the observations themselves illustrate the idea that inheritance is a controlling factor, whereas the modelling supports this idea. It is important to separate the two in order to broaden up the audience.

Line 18 "across a range of geophysically-constrained rift parameters." Perhaps this is not really a relevant point in the conclusions.

Manuscript

Line 24 and ff. Sentence is unclear, perhaps rewrite

Line 26-27. The concept of the "segmentation" needs to be explained better. This is, in principle, the fundamental reason why the 2D approach does not work: linear rifts are relatively small segments, while from a regional scale upward, rifts do not look 2D at all. This is a complexity that is ubiquitous, yet remains unaddressed.

Line 28-33. This is important to set the scene, yet I think it needs some rewriting. The focus here is not the modelling in 2D, but our understanding. We do understand that the initial conditions (inherited structures) play a relevant role, yet how these work in a realistic three-dimensional setting has not been addressed.

Line 34-38. Unclear, please rewrite.

Line 39-40. Perhaps, before moving onto the modelling, it can be helpful to describe more in detail the features of the geological case that is modelled. Then, the "Here, we..." section can be separated.

Line 43-46. Perhaps this is too generic. Many details are provided below, so this part could be used for a better purpose.

Line 113-115. This sentence is too generic.

Line 116. I would be careful with geothermal gradients. The observed are the result of the rifting, they cannot be easily inferred to be initial conditions, can they?

Line 134-136. I would recommend to expand the comparisons. Because the geological features are described way ahead, the comparisons end up too late in the paper and what exactly is the geological observation and how well this is match remains unclear.

Line 137-150. These paragraphs read like a summary. I suppose these would be better used to discuss what exactly is matched and what is not, discuss briefly the mismatch and limitations and, most importantly, concluding with examples of how the outcomes are relevant. Segmentations like the one modelled are global features, I am pretty sure there are other parts of the world where lateral heterogeneities are present and can be deemed causes for segmentation. Perhaps this could widen the audience.

The figures need a rethinking. I have provided above some indications.

Response to the referees

Reviewer #1 (Remarks to the Author):

Dear Authors,

I have now read your paper titled: “Rheological inheritance controls the formation of segmented rifted margins in cratonic lithosphere”. I think that the paper is generally well written and on a very relevant topic. The combination of 2D and 3D models works well. The quality of the figures is also very high. Overall, I think that the paper should eventually be published following the revisions outlined below. My main concerns relate to the presentation of the work in the context of very similar previous studies and also the justification of the study area. In addition, I note a number of minor points that should also be considered.

First, we would like to thank the reviewer for his positive assessment of our work and constructive comments and suggestions, which helped us make some major improvement to the work presented in the first version of the manuscript.

All the points raised by the reviewer were taken into account in our revision and are discussed in detail below. But we want to provide a brief response to the two main points regarding:

1- Similarity with previous studies:

Indeed, many studies have talked the role of inheritance during lithospheric deformation, but all of them, including the ones cited by the reviewer, focussed of structural inheritance – i.e., pre-existing sutures/shear zones (e.g., Heron et al., 2019; Santimano & Pysklywec, 2020). Our work, however, is aimed at examining the role of inherited changes in the thickness, composition, and temperature of the lithosphere. These three elements control the initial rheology of the lithosphere, which we believe in the case of the Labrador Sea is changing along the margin and between the different margin segments as a result of the Archean and Proterozoic tectonic evolution. Many published studies in the 2000s, by Huisman and Beaumont for instance, used 2-D geodynamic modelling to instigate how rheology controls rifting and rifted margin geometry. But to date, and as far as we are aware, none of the published literature looked at pre-rift changes in the rheology of the lithosphere. The novelty of our work is the 3-D geodynamic modelling setup, which incorporate regional changes in the initial rheology of the lithosphere.

2- The justification of the study area:

The Labrador Sea opened across different terrains of varying ages and tectonic history. These terrains continue onshore, where they are part of the Eastern margin of the Canadian Shield. In addition, various types of geophysical studies examined the nature of the crust and mantle lithosphere of the Canadian Shield, which provide a relatively good understanding of the pre-rift/initial properties of the lithosphere. This was not clear in our first manuscript and now is discussed in the introduction and the discussion sections.

Major points:

1) Previous work 3D modelling the NW Atlantic: Heron et al. (2019), Santimano & Pysklywec (2020), Clarke & Beutel (2019) as well as other numerical modelling studies.

The study by Heron et al. (2019) uses a very similar 3D numerical modelling approach to investigate very similar processes primarily focused on the Davis Strait region but also including the Labrador Sea and Baffin Bay (i.e. the whole NW Atlantic - as defined by Abdelmalak et al. 2018). Both the present study and Heron et al. (2019) use the ASPECT numerical model and both are investigating tectonic inheritance and margin segmentation. In addition, Santimano & Pysklywec (2020) also used 3D numerical models to investigate the NW Atlantic, albeit slightly to the north, but still relevant to the present study. Whilst Clarke & Beutel (2019) also conducted 3D numerical modelling experiments of the region looking at stress concentration and implications for magmatism. I therefore think it is not correct when the authors state that “to date no studies have explicitly examined the effects of variable lithospheric structure on the 3D evolution of continental rifting and rifted margin segmentation”. Not only have such models been produced elsewhere (e.g. Koopmann et al., 2014), but similar models have actually already been produced for the present study area. However, I don't think this necessarily means that the present study is not sufficiently original because the work of Heron et al. (2019) is focused on the role of mantle structure whereas the present study is focused on shallower inheritance, and the other studies mentioned are focused on different aspects of rift evolution. I do think however that a much greater discussion and integration of the results with these studies, and in particular the work of Heron et al. (2019) is needed. Currently I feel it is a bit disingenuous of the authors to be claiming that this the first study of its kind in the region. To reiterate though, I don't think this is necessarily a reason for rejection, or a major flaw of the study, but it does mean that a substantial change to the manuscript is required.

The four studies mentioned above tackle different aspects of the evolution of rifted margins, and we would like first to state, very briefly, the aim of each one:

- Heron et al. (2019) used numerical modelling to examine the nature of an inherited lithospheric structure (i.e., geometry and depth) between the North Atlantic Craton and the Nagssugtoqidian domain, and its role in the evolution of the Davis Strait and the segmentation between the Labrador Sea and the Baffin Bay.
- Santimano & Pysklywec (2020) used analogue modelling to investigate the influence of lithospheric mantle scars and rheology on the formation of the intraplate orogenies (e.g., Eurekan fold and thrust belt).
- Clarke & Beutel (2019) focussed on the origin (plume vs. plate) of the Palaeocene picritics of the Davis Strait, and used elastic finite-element modelling to test the viability of bringing picritic partial melts rapidly to the surface.
- Koopmann et al. (2014) looked at the role of rift propagation delay in controlling mantle flow beneath the rift axis and the ensuing segmentation in the distribution of extrusive volcanism at volcanic passive margins.

The studies by Heron et al. (2009) and Koopmann et al. (2014) certainly looked at rift segmentation but from the perspective of structural inheritance and along-strike rift propagation, respectively. Whereas, our study focusses mainly on the role of inherited variation in lithospheric thickness, composition, and geothermal gradient - i.e., rheology - on the 3D evolution of continental rifting and rifted margin segmentation.

We acknowledge that our 2D models are comparable to many numerical modelling studies that tested the impact of lithosphere rheology on rift geometry and rifted margin evolution, as discussed in our introduction. However, to our knowledge, none of the published studies, including the ones cited by the reviewer, tested an 3D setup where the initial (pre-rift) lithosphere rheology varies along strike.

That is said, we recognise that our statement that “to date no studies have explicitly examined the effects of variable lithospheric structure on the 3D evolution of continental rifting and rifted margin segmentation”, might be ambiguous and was modified to avoid any confusion.

2) Nature of segmentation on the Labrador margin and elsewhere in the region: A question of scale?

This comment has two parts a) and b).

a) Previous work on the north west Atlantic does indeed note significant rift segmentation (Heron et al., 2019). However, the major segmentation of this rift system is into 3 major regions. From south to north these are: 1) the Labrador Sea, 2) the Davis Strait and 3) Baffin Bay. Of course, it is possible that secondary and smaller scale segmentation of these regions (i.e. the Labrador Sea) exists but this is very much secondary, and it needs to be presented in the context of the larger scale rift segmentation which previous work also links to structural inheritance (e.g., Peace et al., 2018b).

We agree with the reviewer comment that segmentation occurs at different scales. The first order segmentation between the South, central, and North Atlantic is controlled by the geodynamics of plate tectonics. The second order segmentation in the case of the NW Atlantic occurs between the Labrador Sea and the Baffin Bay, through the Davis Strait. Here we are interested in segmentation within a single rift basin that is governed by relatively uniform geodynamic conditions – e.g., the Labrador Sea.

This distinction is now explicitly stated in the revised introduction of the manuscript.

b) Related to the point a), I have some comments on the actual nature of segmentation in the Labrador Sea that I feel need addressing also. Considering the crustal structure of the Labrador sea region as presented by Welford and Hall (2013) following their gravity inversion study, it is not entirely clear to me that significant segmentation of the margin occurs at the crustal scale as implied by the present study. In addition, Peace et al. (2017) present a LAB (lithosphere–asthenosphere boundary) depth model for the entire NW Atlantic (Fig. 5b in Peace et al. 2017). Again, the LAB depth model does not really show significant segmentation on the Labrador margin. As such, I feel that the authors should present their evidence for segmentation of the Labrador margin in the context of this work. Is the evidence for the segmentation based around the different geological terrains that the Labrador Sea bisects? Essentially, I just felt that a better discussion of the nature of the segmentation based on the previous geological and geophysical work in the region was needed.

The segmentation in the Labrador Sea is discussed in detail by Gouiza & Paton (2019) and Keen et al. (2018a & 2018b), who identified three rift segments showing major changes in rift geometry, crustal architecture, timing and magmatic budget of the breakup along the Labrador margin. The findings of these three papers and other geological and geophysical studies in the region are now summarized in the revised introduction. Additional details are also given in the supplementary information (e.g., changes in the amount crustal extension, timing of continental breakup, and seafloor spreading).

Our hypothesis is that the margin segmentation is the result of the variability in **pre-rift** lithospheric structure, composition and geothermal gradient. It is expected that this variability gets diluted during rifting because of the intense thinning of the crust and the mantle lithosphere, which is shown in the current depth models of the Moho and the LAB underneath the offshore basin (e.g., Welford & Hall, 2013, Peace et al., 2017). However, the unstretched onshore and proximal domains

of the margin, still document this variability. The seismic refraction line from Funck et al. (2001) show a ca. 15km thickening of the continental crust across the Grenville Suture. Mareschal and Jupart (2004) document major variations in surface heat flow in the hinterland of the Labrador margin (i.e., Canadian Shield), which is explained by changes in crustal heat production (i.e., changes in crustal thickness and/or crustal composition) and depth to the base of the lithosphere. Other geophysical data from the Canadian Shield also suggests that the Archean terrains and the surrounding Proterozoic to Palaeozoic domains are underlain by a geochemically different upper mantle (Shapiro et al., 2004; Yuan & Romanowicz, 2010).

3) Nature of the SW Greenland margin and incorporation of this into models.

Currently, the present study by Gouiza and Naliboff focuses primarily on the Labrador margin. This is understandable given that much more is known about the Labrador margin than its SW Greenland conjugate. However, an abundance of previous work has concluded that to fully comprehend a rift prior to breakup both conjugates must be considered in models (e.g., Wilson et al., 2001). This is particularly relevant in the Labrador Sea where the structure of the margins has been shown to be asymmetric (Peace et al., 2016). The obvious implication of this asymmetry is the question of whether the apparent segmentation that forms the focus of the study by Gouiza and Naliboff also occurs in SW Greenland? Overall, I think that the variable structure of the two conjugates should be expanded upon more with particular emphasis on the segmentation aspect.

It is true that to fully understand the evolution of a rifted margin, both conjugates must be considered. The reviewer rightly points to the fact that the Canadian side of the Labrador Sea is much more studied than the Greenland side. However, the magmatic and structural asymmetry documented between the two conjugates suggests a rifting that is dominated by simple shear and asymmetric continental breakup, with the Labrador margin is the lower plate and the SW Greenland margin is the upper plate (Peace et al., 2016; Gouiza & Paton, 2019). This is consistent with the modelled crustal structure of the Labrador Sea that suggests a much narrower continental crustal along SW Greenland than along the Labrador margin (Welford & Hall, 2013).

4) Previous work specifically on structural inheritance in the NW Atlantic

The present study by Gouiza and Naliboff is essentially modelling structural inheritance at a very large scale. Given this, I felt that further discussion of the work in the region looking at inheritance is needed (e.g., Wilson et al., 2006; Peace et al., 2017, 2018a, 2018b).

There are three main types of inheritance that influence the evolution of plate tectonics, namely structural, compositional and thermal inheritance (e.g., Manatschal et al., 2015). In this work, we focus on the compositional and thermal inheritance (i.e., rheological inheritance) rather than the structural one. This point is now explicitly stated in the revised introduction.

5) Choice of the study area

The authors imply that the Labrador Sea is one of the only segmented rifts globally (Line 39). However, this is clearly not the case as many very recent studies are focused on this phenomena (Heron et al., 2019; Scholz et al., 2020). I felt when reading the text that the authors should better justify the choice of study area.

We agree with the reviewer that the Labrador Sea is not the only segmented rifted margin. Our sentence refers to the preceding paragraph where we mention that the lack of sufficient subsurface

data prevents an accurate assessment of the extent of segmentation in many rifted margins and hinders the reconstruction of any along-strike structural and crustal variability. In the case of the Labrador margin, a wide range of geological and geophysical constraints are available to inform the numerical design and the assessment of the modelling results.

The justification of the study area choice is now better explained in the revised introduction (lines 40-44)

6) Mantle Plumes?

Given that the models presented by Gouiza and Naliboff include variable thermal conditions as a starting point, I am wondering if any inferences can be made about the hypothesised mantle plume in the region (e.g., Gerlings et al., 2009)? This may not be applicable, but I am wondering if any of the features sometime attributed to a plume (e.g., regional breakup) occur in the models in the absence of a plume?

Unfortunately, the visco-plastic material model used here doesn't allow for melting to be incorporated. However, given the ongoing debate about the process behind the volcanics found in the Davis Strait and the northern Labrador Sea that was raised by the reviewer, we added a new section (lines 147-175) about the potential of decompression melting based on our 3D modelling results.

we used a post-processing script to calculate decompression melting of mantle material using the formulation of Katz et al. (2003). Our calculations show that the interplay between extension rate and mantle geochemistry could be the main factor controlling the magmatic budget during rifting and continental breakup in the Labrador Sea, without the need of excess heat from a plume.

Minor points:

Title: Why doesn't the title mention the Labrador Sea when the whole paper is focused on this region?

It is true that we use the Labrador Sea as a case study, but we believe that our findings apply to rift and rifted margins in general, which usually form on a rheologically heterogeneous (including cratonic) lithosphere. As the reviewer mentioned above, rift segmentation is documented in many rifted margins and we adopted a title that can attract a wider audience regardless of their geographic area of research.

Lines 11-12: I am not sure that margin segmentation is completely enigmatic, lots of previous work links it to inheritance.

The reviewer is right, many studies have proposed a link between inheritance and rift/margin segmentation. We meant enigmatic in the sense that it could be driven by multiple factors, rather than one distinct process.

Correction implemented:

Old: (...) the origin of strong segmentation along the length of margins remains relatively enigmatic and may reflect multiple competing factors.

New (Lines 13-14): (...) the origin of strong segmentation along the length of margins remains relatively ambiguous and may reflect multiple competing factors.

Lines 13-14: I disagree with this statement and I would say that previous studies have made progress here. See for example Heron et al. (2019) which is literally modelling the same region as the present study.

Correction implemented:

Old: Given that rifting frequently initiates on complex tectonics sutures, structural inheritance is frequently invoked as an origin of margin segmentation, although to date no studies have clearly elucidated the link between inheritance and 3D rifted margin geometries.

New (lines 14-16): Given that rifting frequently initiates on heterogenous basements with a complex tectonic history, structural inheritance is frequently examined, but the link between inherited compositional and rheological heterogeneities and 3-D rifted margin geometries is often overlooked.

Line 15: “thermal-mechanical modelling” – might be worth mentioning ASPECT here?

We don't think it is necessary to mention ASPECT in the abstract and since the referee is not considering this to be a major and essential correction, no change was implemented.

Line 15: “continental rifting” – And breakup?

Correction implemented:

Old: Here, we use recent observations from the Labrador Sea and thermal-mechanical simulations of continental rifting to constrain the effects of inherited variable lithospheric properties on margin segmentation.

New (lines 16-18): Here, we use the ASPECT code to build thermo-mechanical simulations of continental rifting, constrained by observations from the Labrador Sea, to unravel the effects of inherited variable lithospheric properties on margin segmentation.

Line 17: “Lithospheric thickness” - Peace et al. (2016) show a figure that implies there is minimal variation in LAB (lithosphere-asphenspehre Boundary) depth along the margins of the Labrador Sea. The major structure is associated with the Davis Strait to the North. In addition, the crustal thickness from gravity inversion shown in Welford and Hall (2013) shows minimal variation along the Labrador margin.

This is in relation to the major comment #2b and as explained above the variability in crustal and lithospheric thicknesses referred to is pre-rift. It is expected that this variability gets diluted because of the intense thinning of the crust and the lithosphere during rifting, as shown in the current depth models of the Moho and the LAB underneath the offshore basin (e.g., Welford & Hall, 2013, Peace et al., 2017). However, the unstretched hinterland and proximal domains of the margin, still document this variability. The seismic refraction line from Funck et al. (2001) show a ca. 15km thickening of the continental crust across the Grenville Suture. Mareschal and Jupart (2004) document major variations in surface heat flow in the hinterland of the Labrador margin (i.e., Canadian Shield), which is explained by changes in crustal heat production (i.e., changes in crustal thickness and/or crustal composition) and depth to the base of the lithosphere.

Correction implemented:

Old: The modelling results demonstrate that N-S variations in lithospheric thickness, crustal structure, and rheology within the pre-rift Canadian Shield produce sharp gradients in rifted margin width and the timing of breakup, leading to strong margin segmentation across a range of geophysically-constrained rift parameters.

New (lines 18-20): The modelling results demonstrate that variations in the initial crustal and lithospheric thickness, composition, and rheology produce sharp gradients in rifted margin width, the timing of breakup and its magmatic budget, leading to strong margin segmentation.

Line 26: “segments” - So the major segmentation of the NW Atlantic region is the Labrador Sea - Davis Strait - Baffin Bay. I think it would help if the authors could justify why they are focusing on a part of this system with less segmentation than elsewhere in the NW Atlantic? (I realise that this is also the subject of one of the more major points above).

This is a matter of scale as discussed above (major comment #2a). The first order segmentation between the South, central, and North Atlantic is controlled by the geodynamics of plate tectonics. The second order segmentation in the case of the NW Atlantic occurs between the Labrador Sea and the Baffin Bay, through the Davis Strait. Here we are interested in segmentation within a single rift basin that is governed by relatively uniform geodynamic conditions – e.g., the Labrador Sea.

Lines 34-35: “Despite these advances, to date no studies have explicitly examined the effects of variable lithospheric structure on the 3D evolution of continental rifting and rifted margin segmentation” – I disagree with this statement.

This point is related to the major comment #1, which is discussed in detail above. We recognise that our statement, although valid, might be ambiguous and was modified to avoid any confusion.

Correction implemented:

Old: Despite these advances, to date no studies have explicitly examined the effects of variable lithospheric structure on the 3D evolution of continental rifting and rifted margin segmentation.

New (lines 38-40): Nonetheless, to date no studies have explicitly examined the effects of a heterogeneous pre-rift lithosphere, with domains of varying rheology (i.e., composition, thickness, and thermal structure), on the 3-D evolution of continental rifting and rifted margin segmentation.

Line 53: But what about the SW Greenland margin?

Correction implemented and a new paragraph added:

New (lines 58-61): The crustal architecture, depicted by seismic data and gravity modelling, suggest an asymmetric rifting in the Labrador Sea and that the line of breakup was closer to the Greenland side (i.e., upper plate) than the Labrador side (i.e., lower plate). It implies that most of the crustal stretching is preserved on the Canadian side of the basin.

Line 61-62: It is not universally accepted that the volcanic rocks in the Davis Strait are plume related (Peace et al., 2017; Clarke and Beutel, 2019).

Correction implemented (and the papers by Peace et al. (2017) and Clarke & Beutel (2019) were cited):

Old: In the latter, breakup appears to coincide with Iceland-plume-related flood basalts (ca. 61–56 Ma) located near the Davis Strait.

New (lines 57-58): In the latter, breakup appears to coincide with flood basalts located near the Davis Strait (ca. 61–56 Ma), whose link to the Iceland plume is suggested but still debated.

Line 66: Again, what about the SW Greenland margin? Can you comment on the structure of this margin?

New (lines 58-61): The crustal architecture, depicted by seismic data and gravity modelling, suggest an asymmetric rifting in the Labrador Sea and that the line of breakup was closer to the Greenland side (i.e., upper plate) than the Labrador side (i.e., lower plate). It implies that most of the crustal stretching is preserved on the Canadian side of the basin.

Line 95: The asymmetric extension is explicitly studied by Peace et al. (2016)

Peace et al. (2016) is now cited in line 60.

Line 137-138: Ok, but so do lots of previous studies. I think it would be good to explicitly state what is different about this study.

This point is related to the major comment #1. Many studies used numerical modelling to examine the role of structural inheritance in rifting and rifted margin evolution. Our study focusses on rheological inheritance rather than structural inheritance.

Correction implemented:

Old: Our geodynamic models emphasise the role of pre-rift lithospheric heterogeneities in defining rift evolution and crustal architecture along rifted margins.

New: Our geodynamic models emphasise the role of pre-rift lithospheric rheological heterogeneities in defining rift evolution and crustal architecture along rifted margins.

Figure 1: This is a very nice figure and gives a good regional overview. My one comment would be that it really does emphasize the “1-sided” approach to studying the Labrador Sea rift and shows how the SW Greenland margin has been sort of ignored. There are openly available crustal scale seismic lines on the SW Greenland margin (Chian et al., 1995) that could be considered here.

Crustal section from the conjugate SW Greenland margin was added to Figure 1 (Chian & Loudon, 1994)

Figure 2: This figure gives a very clear overview of the modelling setup.

This figure was moved to the Supplementary Information, which was suggested by one of the other reviewers.

Figure 3: I found this figure to be a bit hard to get into. I am sure there is a reason but why are the 8 Myr images shown on the left and the 4 Myr images on the right? I appreciate that a lot of different

model iterations are being shown on one figure but I found this figure quite hard to interpret. Can it be separated out into more figures or in that not allowed in the journal?

This Figure was simplified and split into two figures.

The reason behind showing the low extension rate (5 mm/yr) and high extension rate (10 mm/yr) models at x Myr and 2x Myr, respectively, is to allow comparison of different models but with the same amount of extension.

Figure 4: This is a very nice figure and shows off the 3D model nicely.

Minor changes, requested by another reviewer, were introduced.

Kind regards,

Dr. Alexander L. Peace
McMaster University, Canada

References cited in this review

- Abdelmalak, M.M., Planke, S., Polteau, S., Hartz, E.H., Faleide, J.I., Tegner, C., Jerram, D.A., Millett, J.M., and Myklebust, R., 2018, Breakup volcanism and plate tectonics in the NW Atlantic: *Tectonophysics*, v. 760, p. 267–296, doi:10.1016/j.tecto.2018.08.002.
- Chian, D., Loudon, K.E., and Reid, I., 1995, Crustal structure of the Labrador Sea conjugate margin and implications for the formation of nonvolcanic continental margins: *Journal of Geophysical Research*, v. 100, p. 24239, doi:10.1029/95JB02162.
- Clarke, D.B., and Beutel, E.K., 2019, Davis Strait Paleocene Picrites: Products of a Plume or Plates? *Earth-Science Reviews*, doi:10.1016/j.earscirev.2019.01.012.
- Gerlings, J., Funck, T., Jackson, H.R., Loudon, K.E., and Klingelhofer, F., 2009, Seismic evidence for plume-derived volcanism during formation of the continental margin in southern Davis Strait and northern Labrador Sea: *Geophysical Journal International*, v. 176, p. 980–994, doi:10.1111/j.1365-246X.2008.04021.x.
- Heron, P.J., Peace, A.L., McCaffrey, K.J.W., Welford, J.K., Wilson, R.W., van Hunen, J., and Pysklywec, R.N., 2019, Segmentation of rifts through structural inheritance: Creation of the Davis Strait: *Tectonics*, doi:10.1029/2019TC005578.
- Koopmann, H., Brune, S., Franke, D., and Breuer, S., 2014, Linking rift propagation barriers to excess magmatism at volcanic rifted margins: *Geology*, v. 42, p. 1071–1074, doi:10.1130/G36085.1.
- Peace, A.L., Dempsey, E.D., Schiffer, C., Welford, J.K., Ken, J.W., McCaffrey, K., Imber, J., and Phethean, J.J.J., 2018a, Evidence for Basement Reactivation during the Opening of the Labrador Sea from the Makkovik Province, Labrador, Canada: Insights from Field Data and Numerical Models: *Geosciences*, v. 8, p. 308, doi:10.3390/geosciences8080308.
- Peace, A.L., Foulger, G.R., Schiffer, C., and McCaffrey, K.J.W., 2017, Evolution of Labrador Sea–Baffin Bay: Plate or Plume Processes? *Geoscience Canada*, v. 44, doi:10.12789/geocanj.2017.44.120.
- Peace, A., McCaffrey, K.J.W., Imber, J., van Hunen, J., Hobbs, R., and Wilson, R., 2018b, The role of pre-existing structures during rifting, continental breakup and transform system development, offshore West Greenland: *Basin Research*, v. 30, p. 373–394, doi:10.1111/bre.12257.
- Peace, A.L., McCaffrey, K.J.W., Imber, J., Phethean, J., Nowell, G., Gerdes, K., and Dempsey, E., 2016, An evaluation of Mesozoic rift-related magmatism on the margins of the Labrador Sea: Implications for rifting and passive margin asymmetry: *Geosphere*, v. 12, doi:10.1130/GES01341.1.
- Santimano, T., and Pysklywec, R., 2020, The influence of lithospheric mantle scars and rheology on intraplate deformation and orogenesis: Insights from tectonic analogue models: *Tectonics*, p. 1–19,

doi:10.1029/2019tc005841.

Scholz, C., Shillington, D., Wright, L., Accardo, N., Gaherty, J., and Chindandali, P., 2020, Intrarift fault fabric, segmentation, and basin evolution of the Lake Malawi (Nyasa) Rift, East Africa: *Geosphere*, v. 16, p. 1–19, doi:10.1130/GES02228.1.

Welford, J.K., and Hall, J., 2013, Lithospheric structure of the Labrador Sea from constrained 3-D gravity inversion: *Geophysical Journal International*, v. 195, p. 767–784, doi:10.1093/gji/ggt296.

Wilson, R.W., Klint, K.E.S., van Gool, J.A.M., McCaffrey, K.J.W., Holdsworth, R.E., and Chalmers, J.A., 2006, Faults and fractures in central West Greenland: onshore expression of continental break-up and sea-floor spreading in the Labrador–Baffin Bay Sea: *Geological Survey Of Denmark And Greenland Bulletin*, v. 11, p. 185–204,

http://www.dur.ac.uk/k.j.w.mccaffrey/McCaffrey_pdfs/Wilsonetla_GEUS07.pdf.

Wilson, R.C.L., Whitmarsh, R.B., Taylor, B., and Froitzheim, N., 2001, Introduction: the land and sea approach: Geological Society, London, Special Publications, v. 187, p. 1–8, doi:10.1144/GSL.SP.2001.187.01.01.

Reviewer #2 (Remarks to the Author):

Dear authors,

I have read with attention the paper entitled “rheological inheritance controls the formation of segmented rifted margins in cratonic lithosphere” by Gouiza and Naliboff and I think the idea developed in the paper is new and good, the sciences is despite some typo in the parameters table of good quality, the literature cited is adequate and figure and supplementary movies are informative.

We would like to thank the reviewer for taking the time to review our manuscript and for her constructive and extremely helpful feedback, which helped us implement several key improvements.

The paper reads well but I really think it would definitely benefit of being lengthened to better include the geological arguments that are mostly given in the supplementary material.

As in all modeling study that are related to inheritance, the model set up and all the heterogeneities included in the model will for sure control the results that is no surprise so that means that extra care must be given to argument and justify all these heterogeneities.

Additional descriptions of the geological and geophysical constrained were added to the revised introduction.

In that paper main variations are caused by

1/ Change in lithospheric thicknesses that are not sufficiently argued (why these 3 values, are you taking today's thickness ? from which tomographic models ? why should it be the same 100 Myr ago ?)

The initial lithospheric thicknesses are indeed present-day values based on the work by Shapiro et al. (2004) and Yuan & Romanowicz (2010).

Shapiro et al. (2004) estimated lithospheric thickness using surface heat flux as an a priori constraint on inversions of seismic surface wave dispersion data. Their calculations show an increase in lithospheric thickness from ca. 150-200km south of the Grenville Suture to 200-250 km north of the suture zone.

Yuan & Romanowicz (2010) used a full waveform time domain tomographic inversion method to constrain the lithospheric layering in the Canadian shield. They show comparable lithosphere thickness values, in the order of 150km in the Grenville domain and 200km in the Nain domain in the north.

The last major tectonic event that affected the Canadian Shield was the Grenvillian orogeny (1.08–0.97 Ga), which resulted in north-westerly thrusting and widespread plutonism (Gower, 1996). No other tectonic events that could have had a major impact on the LAB have been recorded since then. This is why we believe that the present-day LAB values of each domain, reported from the Canadian shield, are representative of the LAB values prior to the Labrador rifting. That is said, we know that this assumption comes with a large uncertainty, as deep mantle processes could have affected the Lab.

As mentioned above, additional text was added to the revised introduction to better support the choice of model constraints.

2/ Change in upper crust heat production (there are no arguments but this is extremely important effects on the geotherm).

We agree, this point is essential to our modelling. Variations in upper crust radiogenic heat production is supported by the work of Mareschal & Jupart (2004), which attribute the change in surface heat flow observed in the Canadian Shield, especially across the Grenville suture, to both variations in LAB depth (i.e., basal heat) and composition of the crust (i.e., radiogenic production).

As mentioned above, additional text was added to the revised introduction to better support the choice of model constraints. In addition, notes 2 and 3 in the Supplementary Information addresses the question of heat flow.

3/ Justification for changes in crustal thickness is not clear either, at least it is not straight forward in figure 1 and it is not discussed in the supplementary material. After reading all the material I still wonder what is the underplated material in the grenville belt.

The underplating in Grenville domain is documented by seismic refraction (Funck et al., 2001) and was interpreted to be associated with underplating during the Neoproterozoic-Early Cambrian Iapetan rifting.

4/ Also the supplementary material mentions that the lithosphere of the grenville is more depleted but that does not show on figure 2 and it is not mentioned in the main text. Are you using a different mantle lithosphere type on the Southern profile? If so please put a different coloring on figure 2 and clarify which creep law from supplement table is used in the caption of figure 2.

Indeed, we are using a dry olivine flow law for the mantle lithosphere in the southern segment, instead of the wet olivine flow law used in the central and northern segments.

Correction implemented:

The old Figure 2 was moved to the Supplementary Information as requested by another reviewers. Its caption was modified to reflect the change in the mantle lithosphere. This change in the rheology of the mantle lithosphere is represented by the use of two shades of green in the new figures 2 and 3 as well.

I am not saying the parameters chosen are not correct, I just want to emphasize that they are not enough justified in the text.

Our revised introduction was completely re-written to better justify the constraints (i.e., crust thickness, lithosphere thickness, radiogenic heat) used to setup of the modelling experiments. In addition, other details can be found in the the supplementary information.

Also the paper would benefit from a bit more discussion on propagation of crustal and lithospheric break up in the 3D models and how this fits with geological observations and noteworthy enough with the presence and absence of magmatism... The models do not include any Iceland plume, does it means we don't need a plume to explain the opening of the Labrador Sea and the SDR?

The reviewer is correct, we didn't include the Iceland plume in our numerical simulations. Unfortunately, the visco-plastic material model used here doesn't allow for melting to be incorporated. However, given the ongoing debate about the process behind the volcanics found in the Davis Strait and the northern Labrador Sea, we used a post-processing script to calculate

decompression melting of mantle material using the formulation of Katz et al. (2003). The results of this additional work are now presented in the new section "Extension rate and mantle composition govern magmatic budget during breakup" and in the new figure 5.

So think this paper should be published and that if the authors were to add more geological and geophysical arguments for their set up, and a more details discussion on the significance of their results on rift propagation (comparing with other 3D modelling studies) and on the plume / SDR issue in the main text , it will significantly improve its impact.

Below I least minor comments and typo that need to be fixed but did not arm my understanding of the paper because these are clearly typos and I like to thanks the author for this easy to read and informative paper.

Laetitia Le Pourhiet

Minor comments

1/ Your last concluding statement l. 150 should be removed, it is not true.
You have assumed that it is true, but you have not demonstrated it is true with your contribution.

We believe that our initial premise, that the variations in lithospheric thickness, composition, and thermal structure can produce the observed segmentation in the Labrador Sea, is valid as shown by the results of the 3-D modelling

2/ equations l. 215 - 223

please add spaces after variables, check the accents on strain rate should be a dot... make u and g bold faced since they are vectors. I know it is hard typing equations in ms word but really this is ugly and I would be ashamed to be thanked as a reviewer if you leave it like this.

Corrections implemented.

(We are very proud to have you as a reviewer and definitely don't want you to be ashamed of having your name linked to our manuscript ☺)

Please use k not K for thermal conductivity and rather say that you conserve heat rather than advection diffusion equation... or you should write advection diffusion production of heat equation.

Corrections implemented.

Also you don't need to write deviatoric strain rate since you assume that divergence of velocity field is zero, it is just the strain rate.

While we agree it is not implicitly necessary to write "deviatoric strain rate", we wanted to be sure this point is clear for the readers who are less familiar with the numerical methods.

Do not put brackets around friction angle in the yield criteria but make sure sin and cos are not italic but regular these are functions not variables.

Corrections implemented.

Maybe use II subscript rather than eff since it is the second invariant, first being the pressure.

Corrections implemented.

You mention diffusion creep in your table of parameters but there is no description on how you account for it in the description of the rheology.

Description added in the Methods section (lines 235-236).

3/ Supplementary table:

Why do you have 50 kg/m³ difference in density between the asthenosphere and lithosphere at reference temperature? I understand that you consider that it is depleted in some case but not depleted in the others... but if it is not depleted there is no reason for the lithosphere to be less dense at similar temperature if the Mg number is the same

Geochemical evidence suggests that the continental mantle lithosphere beneath cratons may be compositionally depleted in some locations relative to the Asthenosphere (Griffin et al., 2003/ [https://doi.org/10.1016/S0301-9268\(03\)00180-3](https://doi.org/10.1016/S0301-9268(03)00180-3)). While seismic evidence points toward a more complicated two-layer lithospheric mantle structure beneath cratons, here we represent the lithospheric mantle with a single layer that is 50 kg/m³ less buoyant than the asthenosphere at the equivalent temperature. Future work may explore 3D simulations with a two-layer lithospheric mantle density and rheology structure, which previous 2D modelling efforts (Beaumont and Ings, 2011/ <https://doi.org/10.1029/2012JB009203>) explored with depleted continental mantle lithosphere densities 50-80 kg/m³ lower than reference mantle densities.

This explanation was added to the caption of our Supplementary Figure 1m, showing the model setup.

Please add units for the prefactor A should be Pa⁻ⁿ s⁻¹

Units added.

You give heat equation with conductivity and here you list diffusivity. Either you use constant diffusivity or constant conductivity in the code (I don't know actually) but since the density depends on temperature would be good to write the heat conservation equation according to the quantity you consider as constant.

The equation is now expressed in terms of thermal diffusivity (κ).

Thermal expansivity must be 2.5e-5 not +5

Correction implemented.

The table mention 0.25 for strain weakening factor but the text says it varies from 0.5 to 0.25 (actually you write a factor 2 or 4... would be nice if the table is more consistent with the text)

The strain is weakened by a factor of 4. The table and the text were corrected to make them consistent.

Please add references for the value used for heat production in the upper crust and justify them because the geotherm and changing heat production at the surface changes the temperature of the moho and the depth of the 1300°C isotherm, so it is very important in your study.

This is in relation to the major comment #2 raised and discussed above. More justification of the varying heat production was added to the text (lines 68-74). In addition, Supplementary Information Note 3 is dedicated to the thermal structure of the models, where we justify the thermal parameters used in the modelling.

I believe the underplated crust must be $0.02e-6$ not $0.02e-4$ (otherwise we would see a bump in the geotherm)

Yes correct. Correction implemented.

4/ Figure 2 :

Please change the vertical scale in 2a to be compatible with 2b

Figure 2 is now figure 1 of the Supplementary Information. The fact that (a) is in a 3-D perspective and (b) is 2-D makes it a bit difficult to have both at the same vertical scale. But we did our best to make the vertical scales comparable.

Reviewer #3 (Remarks to the Author):

The paper “Rheological inheritance controls the formation of segmented rifted margins in cratonic lithosphere” by Gouiza and Naliboff is an interesting contribution on the evolution of continental rifts into oceanic basins. This relatively-short phase has a relevant control on the formation of a fundamental feature of our planet, the ocean floor. In fact, the complexities of the rifting stage remain inherited in oceanic lithosphere and control the geometry and growth of these basins for very long time. To date, our understanding of this stage is still poor, this is mostly due to the simplification to a two-dimensional problem, originally invoked for conceptual necessity and computational limitation. However, this simplification has become an obvious limitation. This is now overcome, in the paper by Gouiza and Naliboff. Testing how rifts evolve in three-dimension around pre-existing continental heterogeneities is an elegant solution that clearly advance our understanding.

The modelling is neat and clear although the comparisons and presentation should be improved before the paper is published. The writing is a bit too generic in many places and the structure could be improved: If the geological evidence is presented upfront, then there must be a section where the model’s outcomes are compared with the observables. Of course, if so, it is important to avoid repetition.

The figures do not serve well the paper. The figure 3 does not directly compare to the observation in figure 1, so it remains quite difficult to understand how well this suite of models reproduce the Labrador Sea case. Also, figure 4 is very interesting, however, this should be presented in more detail. For instance, cross section through the 3 segments could be shown. At this stage, the 2D models could be deemed redundant, I guess.

Most importantly, the comparisons between the top surface deformation and the geological case is critical, the hand-drawing on the model does not provide much information. Perhaps a strain rate field or something else would be better.

I would recommend to give the figures a thought, it is a short paper and using one figure for the model set up and one for cramming all the 2d models might not be a helpful strategy: while it is perfectly fine for a long paper, in such a letter this reduces the audience and the impact.

Melbourne 25 September 2020

Fabio A. Capitano

First, we would like to thank the reviewer for his excellent comments and suggestions, which helped us improve the initial version of the manuscript. Major changes were implemented to address all points raised, especially regarding the structure of the manuscript and the figures.

All the changes are explained in detail in our answers bellow.

I provide below some more specific comments

Abstract

The opening sentence “Observations from rifted margins reveal that significant structural and crustal heterogeneity develops through the process of continental extension and breakup” is not very clear and misleading: it emphasises the heterogeneities during the rifting, whereas the paper focuses on those pre-existing.

The premise of our work is that the structural and crustal variability that develops during the rifting and breakup processes is controlled by pre-existing heterogeneities, inherited from the pre-rift tectonics evolution of the lithosphere.

Our opening sentence was reformulated to avoid any confusion.

Correction implemented:

Old: Observations from rifted margins reveal that significant structural and crustal heterogeneity develops through the process of continental extension and breakup.

New: Observations from rifted margins reveal that significant structural and crustal variability and segmentation develops through the process of continental extension and breakup.

Line 14 and ff. "Here, we use recent observations from the Labrador Sea and thermal-mechanical simulations of continental rifting to constrain the effects of inherited variable lithospheric properties on margin segmentation. The modelling results demonstrate ..." This is not a recommended way to proceed. In fact, the observations themselves illustrate the idea that inheritance is a controlling factor, whereas the modelling supports this idea. It is important to separate the two in order to broaden up the audience.

The Reviewer is absolutely right. Observations were used to constrain the inheritance, whereas modelling was used to show that rheological inheritance can produce the observed segmentation in the Labrador Sea.

Correction implemented:

Old: Here, we use recent observations from the Labrador Sea and thermal-mechanical simulations of continental rifting and breakup to constrain the effects of inherited variable lithospheric properties on margin segmentation.

New: Here, we build thermal-mechanical simulations of continental rifting, constrained by observations from the Labrador Sea, to show the effects of inherited variable lithospheric properties on margin segmentation.

Line 18 "across a range of geophysically-constrained rift parameters." Perhaps this is not really a relevant point in the conclusions.

Correction implemented:

Old: The modelling results demonstrate that N-S variations in the initial lithospheric thickness, crustal structure, and rheology within the pre-rift Canadian Shield produce sharp gradients in rifted margin width and the timing of breakup, leading to strong margin segmentation across a range of geophysically-constrained rift parameters.

New (lines 18-20): The modelling results demonstrate that variations in the initial crustal and lithospheric thickness, composition, and rheology produce sharp gradients in rifted margin width, the timing of breakup and its magmatic budget, leading to strong margin segmentation.

Manuscript

Line 24 and ff. Sentence is unclear, perhaps rewrite

Correction implemented:

Old: While this sequence produces genetically similar 'domains' from the un-rifted continent to the seafloor, significant heterogeneity still exists between distinct rifted margins and along the length of individual rifted margins that are characterized by distinct segments.

New (lines 25-29): While this sequence produces genetically similar 'domains' from the un-rifted continent to the seafloor (i.e., proximal, necking, and distal domains), significant segmentation can occur either at plate-scale (e.g., South, Central, and North Atlantic segments), between different rifted margins (e.g., between the Labrador Sea and the Baffin Bay through the Davis Strait), or within a single basin (e.g., Labrador Sea).

Line 26-27. The concept of the "segmentation" needs to be explained better. This is, in principle, the fundamental reason why the 2D approach does not work: linear rifts are relatively small segments, while from a regional scale upward, rifts do not look 2D at all. This is a complexity that is ubiquitous, yet remains unaddressed.

New paragraph was added (lines 25-30) to explain the nature of the segmentation that we are discussing in the manuscript:

While this sequence produces genetically similar 'domains' from the un-rifted continent to the seafloor (i.e., proximal, necking, and distal domains), significant segmentation can occur either at plate-scale (e.g., South, Central, and North Atlantic segments), between different rifted margins (e.g., between the Labrador Sea and the Baffin Bay through the Davis Strait), or within a single basin (e.g., Labrador Sea). Thus, changes in rifting style, strain distribution, crustal architecture, timing and nature of continental breakup can develop across distinct rifted margins and along the length of individual rift systems.

Line 28-33. This is important to set the scene, yet I think it needs some rewriting. The focus here is not the modelling in 2D, but our understanding. We do understand that the initial conditions (inherited structures) play a relevant role, yet how these work in a realistic three-dimensional setting has not been addressed.

Correction implemented:

Improved section (lines 36-44): Furthermore, 3-D numerical simulations can now achieve similar spatial resolutions to 2-D studies and were used to illustrate the margin-parallel effects of structural inheritance, fault network coalescence and out-of-plane or obliqueboundary conditions. Nonetheless, to date no studies have explicitly examined the effects of a heterogeneous pre-rift lithosphere, with domains of varying rheology (i.e., composition, thickness, and thermal structure), on the 3-D evolution of continental rifting and rifted margin segmentation. This in part reflects that many rifted margins initiate on complex pre-rift lithosphere⁶, which may be difficult to accurately reconstruct without sufficient data to connect onshore and offshore domains. In the case of the Labrador Sea, geological and geophysical data indicate an offshore along-strike segmentation that is clearly defined by onshore variations of crustal and lithospheric thickness, composition, and thermal structure.

Line 34-38. Unclear, please rewrite.

Correction implemented:

Old: Despite these advances, to date no studies have explicitly examined the effects of variable lithospheric structure on the 3D evolution of continental rifting and rifted margin segmentation. This

in part reflects that the many rifted margins initiate on complex pre-rift lithosphere, which may be difficult to accurately reconstruct and include short-wavelength structural variations (i.e., brittle faults, ductile shear zones) whose relationship to extensional structures is often difficult to constrain without high-resolution seismic data to connect onshore and offshore structures.

New (lines 38-44): Nonetheless, to date no studies have explicitly examined the effects of a heterogeneous pre-rift lithosphere, with domains of varying rheology (i.e., composition, thickness, and thermal structure), on the 3-D evolution of continental rifting and rifted margin segmentation. This in part reflects that many rifted margins initiate on complex pre-rift lithosphere, which may be difficult to accurately reconstruct without sufficient data to connect onshore and offshore domains. In the case of the Labrador Sea, geological and geophysical data indicate an offshore along-strike segmentation that is clearly defined by onshore variations of crustal and lithospheric thickness, composition, and thermal structure.

line 39-40. Perhaps, before moving onto the modelling, it can be helpful to describe more in detail the features of the geological case that is modelled. Then, the “Here, we...” section can be separated.

The revised introduction now includes more details about the geological constraints extracted from the Labrador hinterland. See lines 45 to 74.

Line 43-46. Perhaps this is too generic. Many details are provided below, so this part could be used for a better purpose.

We think that it is important to highlight the fact that our analysis of the modelling results is informed by observations from the Labrador margin. No change implemented.

Line 113-115. This sentence is too generic.

This sentence was removed.

Line 116. I would be careful with geothermal gradients. The observed are the result of the rifting, they cannot be easily inferred to be initial conditions, can they?

This is now discussed in the introduction (and the Supplementary Information Note 3):

New paragraph (lines 68-74): The observed lithosphere structure in the Labrador Sea and its hinterland is consistent with surface heat flow data, which show an E-W trend within the Labrador Sea, parallel to the rifted margin, and a N-S trend in the hinterland (i.e., Canadian Shield), parallel to the pre-rift tectonic domains. The E-W trend is related to the Mesozoic rifting as heat flow steadily decreases from the distal margin to its hinterland. Whereas, the N-S trends is ascribed to pre-rift variation in both basal heat (i.e., lithosphere thickness) and crustal radiogenic heat production⁴⁰. This suggests that the effect of the Labrador rifting on the thermal structure was restricted to the rifted domain and negligible in the margin hinterland.

Line 134-136. I would recommend to expand the comparisons. Because the geological features are described way ahead, the comparisons end up too late in the paper and what exactly is the geological observation and how well this is match remains unclear.

This is a short paper and we wanted to avoid repeating the detailed geological descriptions that are now in the revised introduction of the manuscript. That is said, we added some details to the

discussion, highlighting the key features (lines 180-191).

Line 137-150. These paragraphs read like a summary. I suppose these would be better used to discuss what exactly is matched and what is not, discuss briefly the mismatch and limitations and, most importantly, concluding with examples of how the outcomes are relevant. Segmentations like the one modelled are global features, I am pretty sure there are other parts of the world where lateral heterogeneities are present and can be deemed causes for segmentation. Perhaps this could widen the audience.

The discussion section was revised according to the reviewer comment.

The figures need a rethinking. I have provided above some indications.
Figures have been updated.

REVIEWER COMMENTS

Reviewer #1 (Remarks to the Author):

Dear authors,

I can confirm that the vast majority of the suggestions I made have been made or otherwise justified. The only one I would have liked to have seen implemented that wasn't is the title which in my opinion does not really reflect the content very well. In the responses letter it is suggested that the title reflects the fact that the findings are more widely applicable. However, the vast majority of the paper is specific to the NW Atlantic. I appreciate that this is just my opinion though and leave this up to the authors and editor to figure out.

The above point should not stop the paper being published though and I look forward to seeing the final paper published, and hope that the authors found my review to be helpful.

Kind regards,

Alex Peace

Reviewer #2 (Remarks to the Author):

The authors have clearly addressed all my concern, the manuscript has substantially improved in quality and I recommend the publication of the ms as it is.

Reviewer #3 (Remarks to the Author):

I have read the revised version of "Rheological inheritance controls the formation of segmented rifted margins in cratonic lithosphere", by M. Gouiza and J. Naliboff. This version has much improved, and I am in general happy with the quality of the figures and the text content.

However, I would strongly recommend tightening up the text more, before the paper is sent for publication. Some parts are not clear, dangle or do not really address a specific point. I have provided below some comments below on the text, although I recommend an overall careful refining of the text.

With my apologies for late review
all the best

Fabio A. Capitano
Melbourne 12 April 2021

Abstract: "Observations from rifted margins reveal that significant structural and crustal variability develops through the process of continental extension and breakup." This is an example of where the writing could be tightened.

Line 14 perhaps remove "and may reflect multiple competing factors"

Line 14 "frequently" perhaps remove. First, this adds little information, second it is repeated in the same sentence.

Lin 18 and ff. "The modelling results demonstrate that variations in the initial crustal and lithospheric thickness, composition, and rheology produce sharp gradients in rifted margin width, the timing of breakup and its magmatic budget, leading to strong margin segmentation" this conclusion is quite evident. Perhaps the authors want to think how this may change the way we look at rifting. Are these pre-existing structures so important? Do they, alone, help determining the end result of rifting?

Line 22 and ff. "The formation of rifted continental margins occurs through multiple phases of extension with distinct structural, sedimentary, and magmatic characteristics^{1,2}. A synthesis of key features at Atlantic rifted margins² suggests that most rifted margins undergo a similar sequence of deformation phases, which reflect progressive thinning of the continental lithosphere that produces a transition from distributed to highly localized deformation."

This text needs tightening up

Line 27-30. There is a missing step between the segmentation observed and the structures: what is segmented? Only the margin? Or the segmentation is accompanied by a variation in strain style? Is it just an offset or a significant control on the deformation style can be inferred?

Line 31. "Inheritance" is confusing here. First, what is inherited? Second, it already implies a process, an evolution. This is better explained in line 32, perhaps invert.

Line 42-45. Perhaps move the case of Labrador Sea in the next paragraph. Then, it is customary to start the paragraph with an indication of what the paragraph says, e.g. "the Labrador Sea is a remarkable example of inheritance, that is the control of pre-existing structures/geotherm/whatnot on the segmentation/strain/geometry..."

Line 76 "The numerical design"?

Line 78-80. Vague conclusions. I suggest strengthening.

Line 82 and ff. Run-on sentence...

Section starting at 149 could present quantitative assessment, too.

210 and ff. "Our findings show that variations in heat flow, lithospheric and crustal thickness and composition exert a first order control on the evolution of rifted margins, which in most cases initiate on a lithosphere with strong inherited lateral heterogeneities. We also demonstrate that observations from margin hinterland could be used as a proxy to gain insights into the initial (thermal and compositional) state of the lithosphere." These conclusions, as stated before, undersell the value of this work!

Coming to some relevant conclusions of use for a wide audience is the bare minimum a scientific paper should do. I recommend the authors to think broadly what the take home message of this work is.

REVIEWER COMMENTS

Reviewer #1 (Remarks to the Author):

Dear authors,

I can confirm that the vast majority of the suggestions I made have been made or otherwise justified. The only one I would have liked to have seen implemented that wasn't is the title which in my opinion does not really reflect the content very well. In the responses letter it is suggested that the title reflects the fact that the findings are more widely applicable. However, the vast majority of the paper is specific to the NW Atlantic. I appreciate that this is just my opinion though and leave this up to the authors and editor to figure out.

The above point should not stop the paper being published though and I look forward to seeing the final paper published, and hope that the authors found my review to be helpful.

Kind regards,

Alex Peace

Reviewer #2 (Remarks to the Author):

The authors have clearly addressed all my concern, the manuscript has substantially improved in quality and I recommend the publication of the ms as it is.

Reviewers 1 and 2 were happy with the changes made and did not suggest any further corrections.

Reviewer #3 (Remarks to the Author):

I have read the revised version of "Rheological inheritance controls the formation of segmented rifted margins in cratonic lithosphere", by M. Gouiza and J. Naliboff. This version has much improved, and I am in general happy with the quality of the figures and the text content.

However, I would strongly recommend tightening up the text more, before the paper is sent for publication. Some parts are not clear, dangle or do not really address a specific point. I have provided below some comments below on the text, although I recommend an overall careful refining of the text.

With my apologies for late review

all the best

Fabio A. Capitanio

Melbourne 12 April 2021

We would like to thank Fabio for reviewing the revised version of our manuscript. Most of the minor corrections, which were proposed, were implemented as explained below.

Abstract: "Observations from rifted margins reveal that significant structural and crustal variability develops through the process of continental extension and breakup." This is an example of where the writing could be tightened.

New sentence:

Rifted margins show significant structural, crustal, and magmatic variability, which develops during continental extension and breakup.

Line 14 perhaps remove "and may reflect multiple competing factors"

Sentence removed. The new sentence reads:

While a clear link exists between distinct margin structural domains and specific phases of rifting, the origin of this variability and the subsequent segmentation along the length of margins remains relatively ambiguous.

Line 14 “frequently” perhaps remove. First, this adds little information, second it is repeated in the same sentence.

We removed “frequently” from the beginning of the sentence. The new sentence reads:

Given that rifting initiates on heterogenous basements with a complex tectonic history, the role of structural inheritance and shear zone reactivation is frequently examined.

Lin 18 and ff. “The modelling results demonstrate that variations in the initial crustal and lithospheric thickness, composition, and rheology produce sharp gradients in rifted margin width, the timing of breakup and its magmatic budget, leading to strong margin segmentation” this conclusion is quite evident. Perhaps the authors want to think how this may change the way we look at rifting. Are these pre-existing structures so important? Do they, alone, help determining the end result of rifting?

This last part of the abstract was modified to reflect the reviewer’s comment. The new paragraph reads:

The modelling results demonstrate that variations in the initial crustal and lithospheric thickness, composition, and rheology dictate the margin geometry at breakup. They are the primary factors responsible of producing sharp gradients in rifted margin width, the timing of breakup and its magmatic budget, leading to strong margin segmentation.

Line 22 and ff. “he formation of rifted continental margins occurs through multiple phases of extension with distinct structural, sedimentary, and magmatic characteristics^{1,2}. A synthesis of key features at Atlantic rifted margins² suggests that most rifted margins undergo a similar sequence of deformation phases, which reflect progressive thinning of the continental lithosphere that produces a transition from distributed to highly localized deformation.”

This text needs tightening up

Paragraph tightened up and now reads:

The formation of rifted continental margins occurs through multiple phases of extension, which results in distinct structural, sedimentary, and magmatic characteristics. Observations from Atlantic rifted margins² suggests that they often undergo a similar sequence of deformation phases, which reflect progressive thinning of the continental lithosphere that produces a transition from distributed to highly localized deformation.

Line 27-30. There is a missing step between the segmentation observed and the structures: what is segmented? Only the margin? Or the segmentation is accompanied by a variation in strain style? Is it just an offset or a significant control on the deformation style can be inferred?

This is explained in the last sentence of the paragraph:

Thus, changes in rifting style, strain distribution, crustal architecture, timing and nature of continental breakup can develop across distinct rifted margins and along the length of individual rift systems.

Line 31. “Inheritance” is confusing here. First, what is inherited? Second, it already implies a process, an evolution. This is better explained in line 32, perhaps invert.

The first part of the sentence “As most rift basins form along (or near) former orogens”, stipulate that the inheritance we are talking about is related to the orogenic phase preceding rifting.

We don’t think that switching the order of the two sentences is necessary, thus no change was implemented.

Line 42-45. Perhaps move the case of Labrador Sea in the next paragraph. Then, it is customary to start the paragraph with an indication of what the paragraph says, e.g. “the Labrador Sea is a remarkable

example of inheritance, that is the control of pre-existing structures/geotherm/whatnot on the segmentation/strain/geometry...”

We implemented the reviewer’s suggestion and moved the sentence to the following paragraph.

Line 76 “The numerical design”?

“Numerical design” was replaced by “modelling setup”

Line 78-80. Vague conclusions. I suggest strengthening.

New sentences:

Our investigation reveals that inherited variations in lithosphere thickness, thermal structure and composition can reproduce key first-order observations from the Labrador. They promote segmentation in rifts and rifted margins by generating along-strike changes in strain distribution, extension processes, magmatic budget, and timing of continental breakup.

Line 82 and ff. Run-on sentence...

This sentence was split into two sentences:

Thermo-mechanical models were developed, in order to examine the effects of the observed variations in lithospheric thickness, composition, and thermal structure on rift segmentation and margin architecture. The modelling setup assimilate the unique onshore geophysical constraints for each domain of the Labrador Sea (Supplementary Fig. 1).

Section starting at 149 could present quantitative assessment, too.

Total volumes of melt were added:

In the case of a dry depleted mantle (i.e., 0 wt% water), the thermal regime in slow rifting (i.e., 5 mm/yr) remains colder than the anhydrous peridotite solidus⁴⁶, and fails to produce any melt (Fig. 5a), whereas, the fast rifting mode (i.e., 10 mm/yr) is able to generate small volumes of melt (**ca. 1e3 km³**) but very late during the rifting process (Fig. 5a). In the second scenario, which assumes a hydrated enriched mantle (i.e., 0.05 wt% water), much larger melt volumes are produced (**ca. 60e3 km³ at 5 mm/yr and 188e3 km³ at 10 mm/yr**) and melt generation starts in the early rift stage regardless of the extension rate (Fig. 5a-d).

210 and ff. “Our findings show that variations in heat flow, lithospheric and crustal thickness and composition exert a first order control on the evolution of rifted margins, which in most cases initiate on a lithosphere with strong inherited lateral heterogeneities. We also demonstrate that observations from margin hinterland could be used as a proxy to gain insights into the initial (thermal and compositional) state of the lithosphere.” These conclusions, as stated before, undersell the value of this work!

Coming to some relevant conclusions of use for a wide audience is the bare minimum a scientific paper should do. I recommend the authors to think broadly what the take home message of this work is.

The main take home message is presented in the second paragraph of the discussion (lines 190-195):

The segmentation is not only expressed by a reactivation of pre-existing structures (e.g., sutures and shear zones), as shown in previous studies, but can also be driven by a change in the processes controlling rifting within each segment, as indicated by our 3-D models. It manifests laterally in the distribution of strain and rift structures, the variability in crustal architecture, and the change in the timing and magmatic budget of continental breakup. In addition, this magmatic budget variability is more likely to be driven by changes in the extension rate during rifting and/or changes in the geochemistry (i.e., fertility/depletion) of the mantle underneath the rifted plate.

The last concluding sentence is a succinct and simplified version of the abovementioned paragraph.

REVIEWERS' COMMENTS

Reviewer #3 (Remarks to the Author):

I have read the letter and the new version of the paper and I am satisfied with both. I have no objections to publication.

all the best

Fabio